# Calibration of Shared Equilibria in General Sum Partially Observable Markov Games

**Nelson Vadori, Sumitra Ganesh, Prashant Reddy, Manuela Veloso**
J.P. Morgan AI Research
{nelson.n.vadori, sumitra.ganesh, prashant.reddy, manuela.veloso}@jpmorgan.com

## Abstract

Training multi-agent systems (MAS) to achieve realistic equilibria gives us a useful tool to understand and model real-world systems. We consider a general sum partially observable Markov game where agents of different types share a single policy network, conditioned on agent-specific information. This paper aims at i) formally understanding equilibria reached by such agents, and ii) matching emergent phenomena of such equilibria to real-world targets. Parameter sharing with decentralized execution has been introduced as an efficient way to train multiple agents using a single policy network. However, the nature of resulting equilibria reached by such agents has not been yet studied: we introduce the novel concept of *Shared equilibrium* as a symmetric pure Nash equilibrium of a certain Functional Form Game (FFG) and prove convergence to the latter for a certain class of games using self-play. In addition, it is important that such equilibria satisfy certain constraints so that MAS are calibrated to real world data for practical use: we solve this problem by introducing a novel dual-Reinforcement Learning based approach that fits emergent behaviors of agents in a Shared equilibrium to externally-specified targets, and apply our methods to a $n$-player market example. We do so by calibrating parameters governing distributions of agent types rather than individual agents, which allows both behavior differentiation among agents and coherent scaling of the shared policy network to multiple agents.

## 1 Introduction

Multi-agent learning in partially observable settings is a challenging task. When all agents have the same action and observation spaces, the work of [9, 11] has shown that using a single shared policy network across all agents represents an efficient training mechanism. This network takes as input the individual agent observations and outputs individual agent actions, hence the terminology *decentralized execution*. The network is trained by collecting all $n$ agent experiences simultaneously and treating them as distinct sequences of local observations, actions and rewards experienced by the shared policy. Since agents may have different observations at a given point in time, sharing a network still allows different actions across agents. It has also been observed in these works or in [15] that one can include in the agents' individual observations some agent-specific information such as the agent index to further differentiate agents when using the shared policy, thus allowing a certain form of heterogeneity among agents.

This brings the natural question: from a game theoretic standpoint, *what is the nature of potential equilibria learnt by agents using such a shared policy?* We show here that such equilibria are symmetric pure Nash equilibria of a higher level game on the set of stochastic policies, which we call *Shared equilibria*.

The second question that follows from this new concept is then *how can we constrain Shared equilibria so that they match specific externally-specified targets?* The latter is referred to as calibration, where we calibrate input parameters of the multi-agent system (MAS) so as to match externally-specified calibration targets, typically coming from real-world observations on the emergent behaviors of agents and groups of agents. For example, MAS modeling behaviors of people in a city may require that agents in equilibria take the subway no more than some number of times a day in average. Constraints such as those previously described can be achieved by having agents of different nature, or *types*, and optimally balancing those types so as to match the desired targets on the emergent behavior of agents. For example, we may want to optimally balance people living in the suburbs vs. living inside a city so as to match the constraint on taking the subway. Even then, repeating the steps (i) pick a certain set of agent types, and (ii) train agents until equilibrium is reached and record the associated calibration loss, is prohibitively expensive.

We solve this problem by introducing a reinforcement learning (RL) agent (*RL calibrator*) whose goal is to optimally balance types of agents so as to match the calibration target, and crucially who learns jointly with RL agents learning a shared equilibrium, avoiding the issue related to repeating (i)-(ii). The result is *CALSHEQ*, a new dual-RL-based algorithm for calibration of shared equilibria to external targets. *CALSHEQ* further innovates by calibrating parameters governing distributions of agent types (called *supertypes*) rather than individual agents, allowing both behavior differentiation among agents and coherent scaling of the shared policy network to multiple agents.

**Our contributions** are **(1)** we introduce the concept of Shared equilibrium that answers the question on the nature of equilibria reached by agents of possibly different types using a shared policy, and prove convergence to such equilibria using self-play, under certain conditions on the nature of the game. **(2)** we introduce *CALSHEQ*, a novel dual-RL-based algorithm aimed at the calibration of shared equilibria to externally specified targets, that innovates by introducing a RL-based calibrator learning jointly with learning RL agents and optimally picking parameters governing distributions of agent types, and show through experiments that *CALSHEQ* outperforms a Bayesian optimization baseline.

**Related work.** Parameter sharing for agents having the same action and observation spaces has been introduced concurrently in [9, 11], and then applied successfully in the subsequent works [10, 15, 22, 26, 27]. [11] showed that out of the their three proposed approaches, (TRPO-based) parameter sharing was the best performer. Although their work considers a cooperative setting where agents maximize a joint reward, parameter sharing is actually the only method out of their proposed three that doesn't require reward sharing, and we exploit this fact in our work. [14] constitutes an excellent survey of recent work in multi-agent deep RL. The recent paper [29] uses a shared policy for worker agents earning individual rewards and paying tax. There, the RL-based tax planner shares some similarities with our RL calibrator, although our calibrator is responsible for optimally picking agent type distribution rather than public information observable by all agents, and updates its policy on a slower timescale so as to allow equilibria to be reached by the shared policy.

The idea of using RL to calibrate parameters of a system probably goes back to [7], in the context of evolutionary algorithms. As mentioned in the recent work [2], there is currently no consensus on how to calibrate parameters of agent-based models. Most methods studied so far build a surrogate of the MAS [2, 17]. The term "surrogate" is very generic, and could be defined as a model that approximates the mapping between the input parameters and some output metric of the MAS. [17] studies classifier surrogates, and in contrast to the latter and other work on calibration, our work is based on a dual-RL approach where our RL calibrator learns jointly with RL agents learning an equilibrium. In our experiments, we compare our approach to a Bayesian optimization baseline that builds such a surrogate. Inverse RL [8] could be used for calibration, but it aims at recovering unknown rewards from input expert policy: in this work we don't need the latter and assume that rewards are known for each agent type, and that the goal is to find the optimal agent type distribution.

## 2   Shared Equilibria in General Sum Partially Observable Markov Games

**Partially Observable Markov Game setting.** We consider a $n$-player partially observable Markov game [12] where all agents share the same action and state spaces $\mathcal{A}$ and $\mathcal{S}$. We make no specific assumption on the latter spaces unless specifically mentioned, and denote the joint action and state as $\boldsymbol{a_t} := (a_t^{(1)}, ..., a_t^{(n)})$ and $\boldsymbol{s_t} := (s_t^{(1)}, ..., s_t^{(n)})$. We assume that each agent $i$ can only observe its

own states $s_t^{(i)}$ and actions $a_t^{(i)}$, hence the partial observability. To ease notational burden, we use the notation $s_t^{(i)}$ for the agents' states instead of the $o_t^{(i)}$ traditionally used in this context, since in our case the full state $s_t$ is the concatenation of agents' states.

**Agent types and supertypes.** In order to differentiate agents, we assign to each agent $i$ a **supertype** $\Lambda_i \in \mathcal{S}^{\Lambda_i}$, with $\mathbf{\Lambda} := (\Lambda_i)_{i \in [1,n]}$. At the beginning of each episode, agent $i$ is assigned a **type** $\lambda_i \in \mathcal{S}^\lambda$ sampled probabilistically as a function of its supertype, namely $\lambda_i \sim p_{\Lambda_i}$ for some probability density function $p_{\Lambda_i}$, and initial states $s_0^{(i)}$ are sampled independently according to the distribution $\mu_{\lambda_i}^0$. This is formally equivalent to extending agents' state space to $\mathcal{S} \times \mathcal{S}^\lambda$, with a transition kernel that keeps $\lambda_i$ constant throughout an episode and equal to its randomly sampled value at $t = 0$. Supertypes are convenient as they allow to think of agents in terms of distributions of agents, and not at individual level, which allows to scale the simulation in a coherent way. In this sense they can be seen as behavioral templates according to which agents can be cloned. Typically, we create groups of agents who share the same supertype, so that the number of distinct supertypes is typically much less than the number of agents. Note that in [11], it is mentioned that the agent index can be included in its state: this is the special case where the supertype is that number, and the type can only take one value equal to the supertype. In our $n$-player market experiments, supertypes model merchants' probabilities of being connected to customers, as well as parameters driving the distribution of their aversion to risk.

**Rewards, state transition kernel and type-symmetry assumption.** Let $z_t^{(i)} := (s_t^{(i)}, a_t^{(i)}, \lambda_i)$. At each time $t$, agent $i$ receives an individual reward $\mathcal{R}(z_t^{(i)}, z_t^{(-i)})$, where the vector $z_t^{(-i)} := (z_t^{(j)})_{j \neq i}$. The state transition kernel $\mathcal{T} : (\mathcal{S} \times \mathcal{A} \times \mathcal{S}^\lambda)^n \times \mathcal{S}^n \to [0, 1]$ is denoted $\mathcal{T}(z_t, s_t')$, and represents the probability to reach the joint state $s_t'$ conditionally on agents having the joint state-action-type structure $z_t$. We now proceed to making assumptions on the rewards and state transition kernel that we call *type-symmetry*, since they are similar to the anonymity/role-symmetry assumption in [19], which only purpose is to guarantee that the expected reward of an agent in (2) only depends on its supertype $\Lambda_i$. Specifically, we assume that $\mathcal{R}$ is invariant w.r.t. permutations of the $n-1$ entries of its second argument $z_t^{(-i)}$, and that for any permutation $\rho$, we have $\mathcal{T}(z_t^\rho, s_t'^\rho) = \mathcal{T}(z_t, s_t')$, where $z_t^\rho, s_t'^\rho$ are the permuted vectors. In plain words, the latter guaranties that from the point of view of a given agent, all other agents are interchangeable, and that two agents with equal supertypes and policies have the same expected cumulative reward. As in [11], our framework contains no explicit communication among agents.

**Shared policy conditioned on agent type.** In the parameter sharing approach with decentralized execution [11, 15], agents use a common policy $\pi$, which is a probability over individual agent actions $a_t^{(i)}$ given a local state $s_t^{(i)}$. This policy is trained with the experiences of all agents simultaneously, and allows different actions among agents since they have different local states. We innovate by including the agent type $\lambda_i$ in the local states and hence define the shared policy over the extended agent state space $\mathcal{S} \times \mathcal{S}^\lambda$. Denoting $\mathcal{X}$ the space of functions $\mathcal{S} \times \mathcal{S}^\lambda \to \Delta(\mathcal{A})$, where $\Delta(\mathcal{A})$ is the space of probability distributions over actions, we then define:

$$\mathcal{X} := [\mathcal{S} \times \mathcal{S}^\lambda \to \Delta(\mathcal{A})], \quad \pi(a|s, \lambda) := \mathbb{P}\left[a_t^{(i)} \in da | s_t^{(i)} = s, \lambda_i = \lambda\right], \quad \pi \in \mathcal{X}. \quad (1)$$

Note that as often done so in imperfect information games, we can add a hidden variable $h$ in $\pi(a_t^{(i)}|s_t^{(i)}, h_{t-1}^{(i)}, \lambda_i)$ to encode the agent history of observations [11]: to ease notational burden we do not include it in the following, but this is without loss of generality since $h$ can always be encapsulated in the state. Due to our type-symmetry assumptions above and given that agents' initial states are sampled independently according to the distributions $\mu_{\lambda_i}^0$, we see that the expected reward of each agent $i$ only depends on its supertype $\Lambda_i$ and the shared policy $\pi$ (it also depends on other agents' supertypes $\mathbf{\Lambda}_{-i}$ independent of their ordering, but since we work with a fixed supertype profile $\mathbf{\Lambda}$ for now, $\mathbf{\Lambda}_{-i}$ is fixed when $\Lambda_i$ is). We will actually need the following definition, which is slightly more general in that it allows agents $j \neq i$ to use a different policy $\pi_2 \in \mathcal{X}$, where $\gamma \in [0, 1)$ is the discount factor:

$$V_{\Lambda_i}(\pi_1, \pi_2) := \mathbb{E}_{\substack{\lambda_i \sim p_{\Lambda_i}, \ a_t^{(i)} \sim \pi_1(\cdot|\cdot, \lambda_i) \\ \lambda_j \sim p_{\Lambda_j}, \ a_t^{(j)} \sim \pi_2(\cdot|\cdot, \lambda_j)}} \left[\sum_{t=0}^{\infty} \gamma^t \mathcal{R}(z_t^{(i)}, z_t^{(-i)})\right], \quad i \neq j \in [1, n], \ \pi_1, \pi_2 \in \mathcal{X}. \quad (2)$$

$V_{\Lambda_i}(\pi_1, \pi_2)$ is to be interpreted as the expected reward of an agent of supertype $\Lambda_i$ using $\pi_1$, while all other agents are using $\pi_2$. This method of having an agent use $\pi_1$ and all others use $\pi_2$ is mentioned

in [13] under the name "symmetric opponents form approach" (SOFA) in the context of symmetric games. Our game as we formulated it so far is not symmetric since different supertypes get different rewards, however we will see below that we will introduce a symmetrization of the game via the function $\widehat{V}$.

What are then the game theoretic implications of agents of different types using a shared policy? Intuitively, assume 2 players are asked to submit algorithms to play chess that will compete against each other. Starting with the white or dark pawns presents some similarities as it is chess in both cases, but also fundamental differences, hence the algorithms need to be good in all cases, whatever the type (white or dark) assigned by the random coin toss at the start of the game. The 2 players are playing a higher-level game on the space of algorithms that requires the submitted algorithms to be good in all situations. This also means we will consider games where there are "good" strategies, formalized by the concept of extended transitivity in assumption 1, needed in theorem 1.

**Shared policy gradient and the higher-level game $\widehat{V}$.** In the parameter sharing framework, $\pi \equiv \pi_\theta$ is a neural network with weights $\theta$, and the gradient $\nabla_{\theta,B}^{shared}$ according to which the shared policy $\pi_\theta$ is updated (where $B$ is the number of episodes sampled) is computed by collecting all agent experiences simultaneously and treating them as distinct sequences of local states, actions and rewards $s_t^{(i)}, a_t^{(i)}, \mathcal{R}(z_t^{(i)}, \boldsymbol{z_t^{(-i)}})$ experienced by the shared policy [11], yielding the following expression under vanilla policy gradient, similar to the single-agent case:

$$\nabla_{\theta,B}^{shared} = \frac{1}{n}\sum_{i=1}^{n} g_i^B, \quad g_i^B := \frac{1}{B}\sum_{b=1}^{B}\sum_{t=0}^{\infty} \nabla_\theta \ln \pi_\theta\left(a_{t,b}^{(i)}|s_{t,b}^{(i)}, \lambda_{i,b}\right)\sum_{t'=t}^{\infty} \gamma^{t'}\mathcal{R}(z_{t',b}^{(i)}, \boldsymbol{z_{t',b}^{(-i)}}) \quad (3)$$

Note that by the strong law of large numbers, taking $B = +\infty$ in (3) simply amounts to replacing the average by an expectation as in (2) with $\pi_1 = \pi_2 = \pi_\theta$. Proposition 1 is a key observation of this paper and sheds light upon the mechanism underlying parameter sharing in (3): in order to update the shared policy, we **a)** set all agents to use the same policy $\pi_\theta$ and **b)** pick one agent at random and take a step towards improving its individual reward while keeping other agents on $\pi_\theta$: by (4), this yields an unbiased estimate of the gradient $\nabla_{\theta,\infty}^{shared}$. Sampling many agents at random $\alpha \sim U[1,n]$ in order to compute the expectation in (4) will yield a less noisy gradient estimate but will not change its bias. In (4), $\widehat{V}$ is to be interpreted as the utility received by a randomly chosen agent behaving according to $\pi_1$ while all other agents behave according to $\pi_2$.

**Proposition 1.** *For a function $f(\theta_1, \theta_2)$, let $\nabla_{\theta_1} f(\theta_1, \theta_2)$ be the gradient with respect to the first argument, evaluated at $(\theta_1, \theta_2)$. We then have:*

$$\nabla_{\theta,\infty}^{shared} = \nabla_{\theta_1}\widehat{V}(\pi_\theta, \pi_\theta), \quad \widehat{V}(\pi_1, \pi_2) := \mathbb{E}_{\alpha \sim U[1,n]}\left[V_{\Lambda_\alpha}(\pi_1, \pi_2)\right], \quad \pi_1, \pi_2 \in \mathcal{X} \quad (4)$$

*where $\mathbb{E}_{\alpha \sim U[1,n]}$ indicates that the expectation is taken over $\alpha$ random integer in $[1,n]$.*

*Proof.* It is known (although in a slightly different form in [20] or [25] appendix D) that the term $g_i^\infty$ in (3) is nothing else than $\nabla_{\theta_1} V_{\Lambda_i}(\pi_\theta, \pi_\theta)$, that is the sensitivity of the expected reward of an agent of supertype $\Lambda_i$ to changing its policy while all other agents are kept on $\pi_\theta$, cf. (2). The latter can be seen as an extension of the likelihood ratio method to imperfect information games, and allows us to write concisely, using (3):

$$\nabla_{\theta,\infty}^{shared} = \frac{1}{n}\sum_{i=1}^{n} \nabla_{\theta_1} V_{\Lambda_i}(\pi_\theta, \pi_\theta) = \nabla_{\theta_1}\frac{1}{n}\sum_{i=1}^{n} V_{\Lambda_i}(\pi_\theta, \pi_\theta) = \nabla_{\theta_1}\mathbb{E}_{\alpha \sim U[1,n]}\left[V_{\Lambda_\alpha}(\pi_\theta, \pi_\theta)\right] \quad \square$$

**Shared Equilibria.** We remind [6] that a 2-player game is said to be symmetric if the utility received by a player only depends on its own strategy and on its opponent's strategy, but not on the player's identity, and that a pure strategy Nash equilibrium $(\pi_1^*, \pi_2^*)$ is said to be symmetric if $\pi_1^* = \pi_2^*$. For such games, due to symmetry, we call **payoff$(\pi_1, \pi_2)$** the utility received by a player playing $\pi_1$ while the other player plays $\pi_2$.

Equation (4) suggests that the shared policy is a Nash equilibrium of the 2-player symmetric game with payoff $\widehat{V}$, where by our definition of the term "payoff", the first player receives $\widehat{V}(\pi_1, \pi_2)$ while the other receives $\widehat{V}(\pi_2, \pi_1)$. This is because $\nabla_{\theta_1}\widehat{V}(\pi_\theta, \pi_\theta)$ in (4) corresponds to trying to improve the utility of the first player while keeping the second player fixed, starting from the symmetric

point $(\pi_\theta, \pi_\theta)$. If no such improvement is possible, we are facing by definition a symmetric Nash equilibrium, since due to symmetry of the game, no improvement is possible either for the second player starting from the same point $(\pi_\theta, \pi_\theta)$. The game with payoff $\widehat{V}$ can be seen as an abstract game (since the 2 players are not part of the $n$ agents) where each element of the strategy set (that is, every pure strategy) is a policy $\pi \in \mathcal{X}$ defined in (1). This type of game has been introduced in [3] as a *Functional Form Game* (FFG), since pure strategies of these games are stochastic policies themselves (but of the lower-level game among the $n$ agents). This motivates the following definition.

**Definition 1.** *(Shared Equilibrium) A shared (resp. $\epsilon-$shared) equilibrium $\pi^*$ associated to the supertype profile $\mathbf{\Lambda}$ is defined as a pure strategy symmetric Nash (resp. $\epsilon-$Nash) equilibrium $(\pi^*, \pi^*)$ of the 2-player symmetric game with pure strategy set $\mathcal{X}$ and payoff $\widehat{V}$ in (4).*

Note that the previously described mechanism **a)-b)** occurring in parameter sharing is exactly what is defined as **self-play** in [3] (algorithm 2), but for the game of definition 1 with payoff $\widehat{V}$. That is, we repeat the following steps for iterations $n$ (i) set all agents on $\pi_{\theta_n}$ (ii) pick one agent at random and improve its reward according to the gradient update (4), thus finding a new policy $\pi_{\theta_{n+1}}$. The natural question is now *under which conditions do Shared equilibria exist, and can the self-play mechanism in (4) lead to such equilibria?* We know [3] that self-play is related to transitivity in games, so to answer this question, we introduce a new concept of transitivity that we call *extended transitivity* as it constitutes a generalization to 2-player symmetric general sum games of the concept of transitivity for the zero-sum case in [3]. There, such a transitive game has payoff $u(x, y) := t(x) - t(y)$. One can observe that this game satisfies extended transitivity in assumption 1 with $\delta_\epsilon := \epsilon$ and $T(x) := t(x)$. Note also that their monotonic games for which $u(x, y) := \sigma(t(x) - t(y))$ (where $\sigma$ is increasing) satisfy extended transitivity as well with $\delta_\epsilon := \sigma^{(-1)}(\epsilon + \sigma(0))$ and $T(x) := t(x)$.

**Assumption 1.** *(extended transitivity) A 2-player symmetric game with pure strategy set $S$ and payoff $u$ is said to be extended transitive if there exists a bounded function $T$ such that:*

$$\forall \epsilon > 0, \exists \delta_\epsilon > 0 : \forall x, y \in S : \text{ if } u(y, x) - u(x, x) > \epsilon, \text{ then } T(y) - T(x) > \delta_\epsilon.$$

The intuition behind assumption 1 is that $T$ can be seen as the game "skill" that is being learnt whenever a player finds a profitable deviation from playing against itself. It will be required in theorem 1 to prove the existence of shared equilibria, which is the main result of this section. Actually, it will be proved that such equilibria are reached by following self-play previously discussed, thus showing that policy updates based on (4) with per-update improvements of at least $\epsilon$ achieve $\epsilon$-shared equilibria within a finite number of steps. In order to do so, we need definition 2 of a *self-play sequence*, which is nothing else than a rigorous reformulation of the mechanism occurring in self-play [3] (algo 2). For $\epsilon$-shared equilibria, assumption 1 is sufficient, but for shared equilibria, we need the continuity result in lemma 1.

**Definition 2.** *A $(f, \epsilon)$-self-play sequence $(x_n, y_n)_{0 \le n \le 2N}$ of size $0 \le 2N \le +\infty$ generated by $(z_n)_{n \ge 0}$ is a sequence such that for every $n$, $x_{2n} = y_{2n} = z_n$, $(x_{2n+1}, y_{2n+1}) = (z_{n+1}, z_n)$ and $f(x_{2n+1}, y_{2n+1}) > f(x_{2n}, y_{2n}) + \epsilon$.*

**Lemma 1.** *Assume that the rewards $\mathcal{R}$ are bounded, and that $\mathcal{S}$, $\mathcal{A}$ and $\mathcal{S}^\lambda$ are finite. Then $V_{\Lambda_i}$ is continuous on $\mathcal{X} \times \mathcal{X}$ for all $i$, where $\mathcal{X}$ is equipped with the total variation metric.*

**Theorem 1.** *Let $\mathbf{\Lambda}$ be a supertype profile. Assume that the symmetric 2-player game with pure strategy set $\mathcal{X}$ and payoff $\widehat{V}$ is extended transitive. Then, there exists an $\epsilon-$shared equilibrium for every $\epsilon > 0$, which further can be reached within a finite number of steps following a $(\widehat{V}, \epsilon)$-self-play sequence. Further, if $\mathcal{S}$, $\mathcal{A}$ and $\mathcal{S}^\lambda$ are finite and the rewards $\mathcal{R}$ are bounded, then there exists a shared equilibrium. In particular, if $(\pi_{\theta_n})_{n \ge 0}$ is a sequence of policies obtained following the gradient update (4) with $\widehat{V}(\pi_{\theta_{n+1}}, \pi_{\theta_n}) > \widehat{V}(\pi_{\theta_n}, \pi_{\theta_n}) + \epsilon$, then $(\pi_{\theta_n})_{n \ge 0}$ generates a finite $(\widehat{V}, \epsilon)$-self-play sequence and its endpoint $(\pi_\epsilon, \pi_\epsilon)$ is an $\epsilon-$shared equilibrium.*

We should comment on the relationship between our extended transitivity and potential games [21]. In 2-player symmetric exact potential games, the deviation $u(y, z) - u(x, z)$ is equal to that of the so-called potential $P(y, z) - P(x, z)$ for all $x, y, z$. The first observation is that extended transitivity links increments of $u$ and $T$ only when improving from a symmetric point $(x, x)$, which isn't as restrictive as in the case of potential games. This motivates us to briefly introduce the (new, to the best of our knowledge) concept of piecewise potential game below. We start by observing that in the 2-player symmetric case, a potential can be defined as a symmetric function (i.e. $P(x, y) = P(y, x)$)

such that $u(y,z) - u(x,z) = P(y,z) - P(x,z)$ for all $x,y,z$. So in some way, the potential increment plays the role of the "derivative" of $u$. We can generalize this observation to piecewise derivatives: define $x \leq y$ if and only if $u(x,x) \leq u(y,x)$ and $u(x,y) \leq u(y,y)$. We say that the symmetric game with payoff $u$ is piecewise potential if there exists *symmetric* functions $P_1$ and $P_2$ such that $P_1(x,x) = P_2(x,x) =: T(x)$ for all $x$ and:

$$\forall x \leq y: \quad u(y,x) - u(x,x) = P_1(y,x) - P_1(x,x), \quad u(y,y) - u(x,y) = P_2(y,y) - P_2(x,y).$$

Symmetric 2-player potential games satisfy the above definition with $P_1 = P_2$. If $x \leq y$, think of $P_1(y,x) - P_1(x,x) \sim \partial_+ u(x,x)$, the one-sided partial derivative from above (with respect to the first argument) and $P_2(y,y) - P_2(x,y) \sim \partial_- u(y,y)$, the one-sided partial derivative from below. For example for $x, y \in [0,1]$, take $u(x,y) := h(x+y)g(x-y)$, with $g(z) = g_1(|z|)$ if $z \geq 0$ and $g(z) = g_2(|z|)$ otherwise, with $g_1(0) = g_2(0) = 1$. It can be shown that this game is piecewise potential with $P_j(x,y) = h(x+y)g_j(|x-y|)$ (under some conditions on $g_1, g_2, h$). That being said, a piecewise potential game satisfies for $x \leq y$:

$$T(y) - T(x) = \underbrace{u(y,x) - u(x,x)}_{\text{player 1 deviation}} + \underbrace{u(y,y) - u(x,y)}_{\text{player 2 deviation}} + \underbrace{P_2(x,y) - P_1(x,y)}_{\text{piecewise assumption}}$$

With the above decomposition, assumption 1 states that if $y$ is an improvement for player 1 starting from $(x,x)$, then for $T(y) - T(x)$ to be large enough, we need the sum of the piecewise term and player 2's deviation to not be too negative, in which case such a game would be extended transitive.

## 3 Calibration of Shared Equilibria

We now turn to the question of calibration, that is of acting on the supertype profile $\mathbf{\Lambda}$ so as to match externally specified targets on the shared equilibrium. In a game that satisfies the conditions of theorem 1, agents will reach a shared equilibrium associated to $\mathbf{\Lambda}$. For the MAS to accurately model a given real world system, we would like the emergent behavior of agents in that equilibrium to satisfy certain constraints. For example, in the $n-$player market setting of section 4, one may want certain market participants to average a certain share of the total market in terms of quantity of goods exchanged, or to only receive certain specific quantities of these goods. The difficulty is that for every choice of $\mathbf{\Lambda}$, one should in principle train agents until equilibrium is reached and record the associated calibration loss, and repeat this process until the loss is small enough, which is prohibitively expensive. The baseline we consider in our experiments follows this philosophy by periodically trying new $\mathbf{\Lambda}$ obtained via Bayesian optimization (BO). One issue is that BO can potentially perform large moves in the supertype space, hence changing $\mathbf{\Lambda}$ too often could prevent the shared policy to correctly learn an equilibrium since it would not be given sufficient time to adapt.

Our solution is therefore to smoothly vary $\mathbf{\Lambda}$ during training: we introduce a RL calibrator agent with a stochastic policy, whose goal is to optimally pick $\mathbf{\Lambda}$ and who learns jointly with RL agents learning a shared equilibrium, but under a slower timescale. The two-timescale stochastic approximation framework is widely used in RL [5, 16] and is well-suited to our problem as it allows the RL calibrator's policy to be updated more slowly than the agents' shared policy, yet simultaneously, thus giving enough time to agents to approximately reach an equilibrium. This RL-based formulation allows us to further exploit smoothness properties of specific RL algorithms such as PPO [24], where a KL penalty controls the policy update. Since the calibrator's policy is stochastic, this is a distributional approach (in that at every training iteration, you have a distribution of supertype profiles rather than a fixed one) which will contribute to further smooth the objective function in (5), cf. [23]. Note that our formulation of section 2 is general enough to accommodate the case where $\mathbf{\Lambda}$ is a distribution $f(\mathbf{\Lambda})$ over supertype profiles: indeed, define new supertypes $\widetilde{\Lambda_i} := f_i$, where $f_1$ is the marginal distribution of $\Lambda_1$, and for $i \geq 2$, $f_i$ is the distribution of $\Lambda_i$ conditional on $(\Lambda_k)_{k \leq i-1}$ (induced by $f$). This means that instead of a fixed $\mathbf{\Lambda}$, one can choose a distribution $f(\mathbf{\Lambda})$ by simply defining supertypes appropriately, and in that case it is important to see that the shared policy at equilibrium will depend on the distribution $f$ rather than on a fixed supertype profile.

The RL calibrator's **state** is the current supertype $\mathbf{\Lambda}$, and its **action** is a vector of increments $\boldsymbol{\delta}\mathbf{\Lambda}$ to apply to the supertypes, resulting in new supertypes $\mathbf{\Lambda} + \boldsymbol{\delta}\mathbf{\Lambda}$, where we assume that $\Lambda_i$ takes value in some subset of $\mathbb{R}^d$. This approach is in line with the literature on "learning to learn" [1, 18], since the goal of the RL calibrator is to learn optimal directions to take in the supertype space, given a

current location. The RL calibrator has full knowledge of the information across agents and is given $K$ externally specified targets $f_*^{(k)} \in \mathbb{R}$ for functions of the form $f_{cal}^{(k)}((z_t)_{t \geq 0})$. Its **reward** $r^{cal}$ will then be a weighted sum of (the inverse of) losses $\ell_k$, $r^{cal} = \sum_{k=1}^{K} w_k \ell_k^{-1}(f_*^{(k)} - f_{cal}^{(k)}((z_t)_{t \geq 0}))$. The result is algorithm 1, where at stage $m = 1$, the supertype profile $\boldsymbol{\Lambda_1}$ is sampled across episodes $b$ as $\boldsymbol{\Lambda_1^b} \sim \boldsymbol{\Lambda_0} + \boldsymbol{\delta\Lambda^b}$, with $\boldsymbol{\delta\Lambda^b} \sim \pi_1^\Lambda(\cdot|\boldsymbol{\Lambda_0})$ and where we denote $\widetilde{\pi}_1^\Lambda := \boldsymbol{\Lambda_0} + \pi_1^\Lambda(\cdot|\boldsymbol{\Lambda_0})$ the resulting distribution of $\boldsymbol{\Lambda_1}$. Then, we run multi-agent episodes $b$ according to (2), each one of them with its supertype profile $\boldsymbol{\Lambda_1^b}$, and record the reward $r_b^{cal}$, thus corresponding to the calibrator state $\boldsymbol{\Lambda_0}$, and action $\boldsymbol{\delta\Lambda^b}$. The process is repeated, yielding for each episode $b$ at stage $m \geq 2$, $\boldsymbol{\Lambda_m^b} \sim \boldsymbol{\Lambda_{m-1}^b} + \pi_m^\Lambda(\cdot|\boldsymbol{\Lambda_{m-1}^b})$, resulting in a distribution $\widetilde{\pi}_m^\Lambda$ for $\boldsymbol{\Lambda_m}$, empirically observed through the sampled $\{\boldsymbol{\Lambda_m^b}\}_{b=1..B}$. As a result, the calibrator's policy $\pi^\Lambda$ optimizes the following objective at stage $m$:

$$V_{\pi_m}^{calib}(\pi_m^\Lambda) := \mathbb{E}_{\boldsymbol{\Lambda} \sim \widetilde{\pi}_{m-1}^\Lambda, \, \boldsymbol{\Lambda'} \sim \pi_m^\Lambda(\cdot|\boldsymbol{\Lambda})+\boldsymbol{\Lambda}, \, \lambda_i \sim p_{\Lambda_i'}, \, a_t^{(i)} \sim \pi_m(\cdot|\cdot,\lambda_i)} \left[ r^{cal} \right] \quad (5)$$

---

**Algorithm 1 (CALSHEQ) Calibration of Shared Equilibria**

**Input:** learning rates $(\beta_m^{cal})$, $(\beta_m^{shared})$ satisfying assumption 2, initial calibrator and shared policies $\pi_0^\Lambda$, $\pi_0$, initial supertype profile $\boldsymbol{\Lambda_0^b} = \boldsymbol{\Lambda_0}$ across episodes $b \in [1, B]$.

1: **while** $\pi_m^\Lambda$, $\pi_m$ not converged **do**
2:     **for** each episode $b \in [1, B]$ **do**
3:         Sample supertype increment $\boldsymbol{\delta\Lambda^b} \sim \pi_m^\Lambda(\cdot|\boldsymbol{\Lambda_{m-1}^b})$ and set $\boldsymbol{\Lambda_m^b} := \boldsymbol{\Lambda_{m-1}^b} + \boldsymbol{\delta\Lambda^b}$
4:         Sample multi-agent episode with supertype profile $\boldsymbol{\Lambda_m^b}$ and shared policy $\pi_m$, with
    $\lambda_i \sim p_{\Lambda_{m,i}^b}$, $a_t^{(i)} \sim \pi_m(\cdot|\cdot, \lambda_i)$, $i \in [1, n]$ cf. (2)
5:     update $\pi_m$ with learning rate $\beta_m^{shared}$ based on gradient (4) and episodes $b \in [1, B]$
6:     update $\pi_m^\Lambda$ with learning rate $\beta_m^{cal}$ based on gradient associated to (5) with episodes $b \in [1, B]$

---

**Assumption 2.** *The learning rates $(\beta_m^{cal})$, $(\beta_m^{shared})$ satisfy $\frac{\beta_m^{cal}}{\beta_m^{shared}} \overset{m \to +\infty}{\to} 0$, as well as the Robbins-Monro conditions, that is their respective sum is infinite, and the sum of their squares is finite.*

If we make the approximation that the distribution $\widetilde{\pi}_m^\Lambda$ is mostly driven by $\pi_m^\Lambda$, then updating the calibrator's policy $\pi_m^\Lambda$ more slowly will give enough time to the shared policy for equilibria to be approximately reached (since $\pi_m^\Lambda$ is a distribution conditional on the previous supertype location, it is reasonable to assume that as learning progresses, it will compensate the latter and yield a distribution $\widetilde{\pi}_m^\Lambda$ approximately independent from $\widetilde{\pi}_{m-1}^\Lambda$). This is reflected in assumption 2, standard under the two-timescale framework, which ensures [28] that $\pi^\Lambda$ is seen as "quasi-static" with respect to $\pi$, and thus that $\pi_m$ in (5) can be considered as having converged to a shared equilibrium depending on $\pi_m^\Lambda$. $\pi^\Lambda$ is then updated based on (5) using a classical single-agent RL gradient update. This process ensures that $\pi^\Lambda$ is updated smoothly during training and learns optimal directions to take in the supertype space, benefiting from the multiple locations $\boldsymbol{\Lambda_m^b}$ experienced across episodes and over training iterations. Our framework shares some similarities with the work on learning to optimize in swarms of particles [4], since at each stage $m$, we have a distribution of supertype profiles empirically observed through the $B$ episodes, where each $\boldsymbol{\Lambda_m^b}$ can be seen as a particle.

## 4   Experiments in a n-player market setting

**Experimental setting.** We conduct experiments in a $n$-player market setting where merchant agents buy/sell goods from/to customers. Merchants try to attract customers and earn income by offering attractive prices to buy/sell their goods, and a merchant $i$ cannot observe prices offered by other merchants to customers, hence the partial observability. We consider **2 distinct supertypes** for **5-10 merchant agents** (merchant 1 is assigned supertype 1 and the $n - 1$ others are assigned supertype 2), which are respectively vectors of size 12 and 11, resulting in **23 parameters** to calibrate in total. For each supertype we have i) 10 probabilities to be connected to 10 clusters of 50 customers each (500 customers in total) corresponding to transactions of various quantities, ii) the merchant's tolerance to holding a large inventory of goods (inventory tolerance - impacting its reward function), which is 1

parameter for supertype 1, and the mean/variance of a normal random variable for supertype 2. In contrast, experimental results in [17] only calibrate 8 or 12 parameters (although not in a RL context). The **calibration targets** we consider are related to i) the fraction of total customer transactions that a merchant can attract (*market share*) and ii) the distribution of individual transactions that a given merchant receives (the constraint is on 9 percentiles of the distribution, for each supertype). All policies are trained with PPO [24], with a KL penalty to control policy updates.

**Baseline.** There is currently no consensus on how to calibrate parameters of agent-based models [2], but existing literature suggests using surrogate-based methods [17]. The baseline we consider here is Bayesian optimization (BO), a method that has been used for hyperparameter optimization. The latter can be considered as similar to this calibration task, and BO will allow us to periodically record the calibration loss related to a certain choice of supertype $\Lambda$, and suggest an optimal point to try next, via building a Gaussian Process-based surrogate of the MAS.

**Performance metrics.** We study 5 experimental settings fully described in supplementary, and evaluate our findings according to the following three criteria 1) calibrator reward in (5), quantifying the accuracy of the equilibrium fit to the target(s) (1=perfect fit), 2) convergence of merchant agents' rewards to an equilibrium and 3) smoothness of the supertype profile $\Lambda$ as a function of training iterations, ensuring that equilibria is given sufficient time to be reached, cf. discussion in section 3.

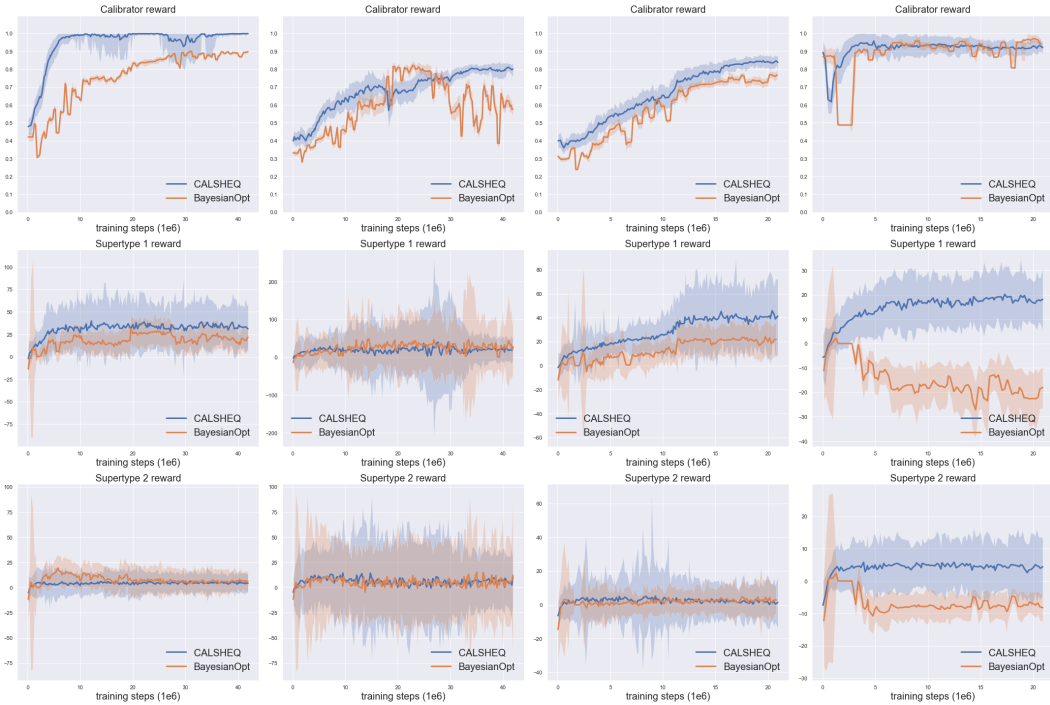

Figure 1: Rewards during training, averaged over episodes $B$ - Calibrator **(Top)**, Supertypes 1/2 **(Mid/Bottom)** - experiments 1-2-3-4. *CALSHEQ* (ours) and baseline (Bayesian optimization). Shaded area represents $\pm 1$ stDev.

**Results.** In figure 1 we display calibrator and agents' reward evolution during training. It is seen that *CALSHEQ* outperforms BO in that i) the RL calibrator's rewards converge more smoothly and achieve in average better results in less time, ii) in experiment 4, supertype 1's reward in the BO case converges to a negative value, which should not happen as merchants always have the possibility to earn zero income by doing nothing. The reason for it is that as mentioned in section 3, BO can potentially perform large moves in the supertype space when searching for a solution, and consequently merchants may not be given sufficient time to adapt to new supertype profiles $\Lambda$. This fact is further seen in figure 3 where we show supertype parameters during training (connectivity and inventory tolerance). It is seen that *CALSHEQ* smoothly varies these parameters, giving enough time to merchant agents on the shared policy to adapt, and preventing rewards to diverge as previously discussed.

The RL calibrator's total reward in (5) is computed as weighted sum of various sub-objectives. In figure 2, we zoom on some of these individual components that constitute the overall reward, together with the associated externally specified target values (figures related to all sub-objectives of all experiments are in the supplementary). It is seen that *CALSHEQ* converges more smoothly and more accurately than BO to the target values. The considered targets are on the percentiles of the distribution of the transaction quantities received by merchants, as well as on the market share. For example, the figure on the top left considers a target of 8 for the 10% percentile of the distribution of transactions received by a merchant of supertype 1.

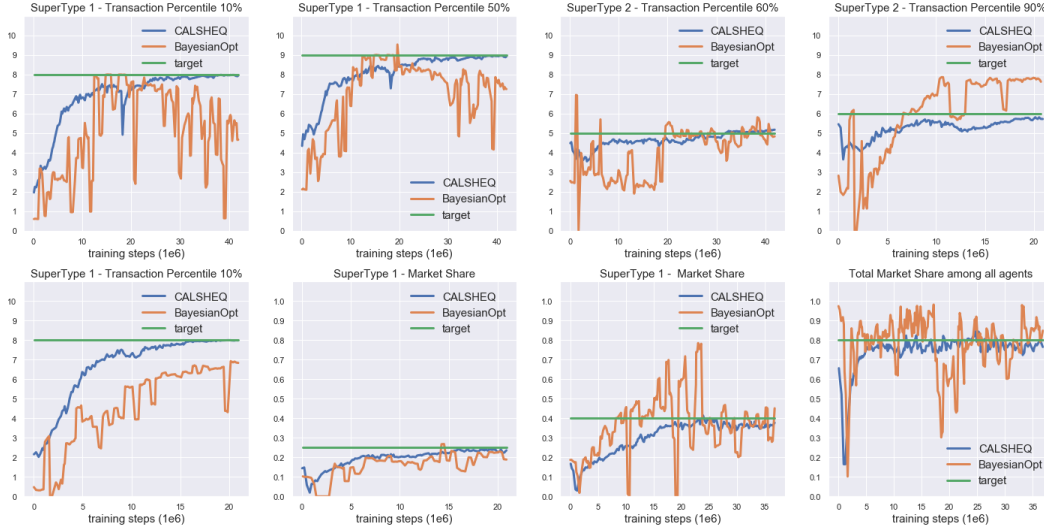

Figure 2: Calibration target fits for transaction quantity distribution percentile and Market Share during training, averaged over episodes $B$ - Experiments 2-2-2-3 **(Top)** and 3-4-5-5 **(Bottom)** - *CALSHEQ* (ours) and baseline (Bayesian optimization).

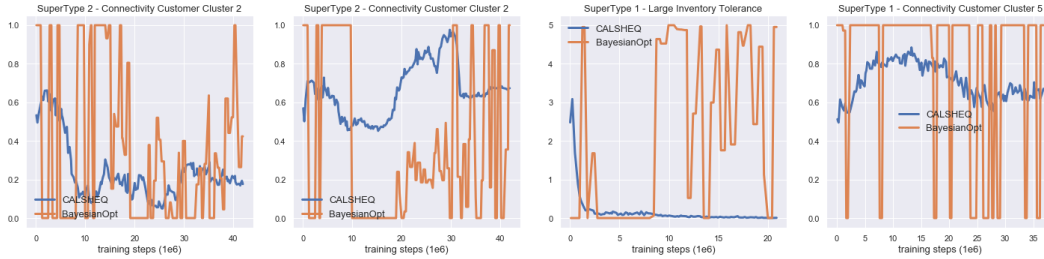

Figure 3: Smoothness of supertype parameters being calibrated during training, averaged over episodes $B$ - Experiments 1-2-4-5 - *CALSHEQ* (ours) and baseline (Bayesian optimization).

## 5 Conclusion

This work was first motivated by the authors wondering what were the game theoretical implications of agents of possibly different types using a shared policy. This led to the concept of Shared equilibrium presented in this paper, which provides insight into the mechanisms underlying policy sharing. From the latter followed the natural question of how to constrain such equilibria by optimally balancing types of agents. The result is a novel dual-RL based algorithm that operates under a two-timescale stochastic approximation framework, *CALSHEQ*, for which we show through experiments that it allows to smoothly bring learning agents to equilibria satisfying certain externally specified constraints. We hope that the present work on calibration will constitute a new baseline and give further ideas to researchers attempting to constrain/calibrate equilibria learnt by learning agents.

## Broader Impact

The first part of our work attempts to bring a formal/theoretical understanding of equilibria reached by agents using a shared policy network and is difficultly applicable to this section.

The second part of our work introduces a novel reinforcement learning based algorithm to calibrate/constrain equilibria learnt by such agents in multi-agent systems/simulators to externally-specified objectives. It is easy to see how this new methodology could be used with fairness targets/objectives in mind, thus leading to more fair and ethical equilibria learnt by reinforcement learning agents. For example, one could constrain the learnt equilibrium to have as few observed non-ethical behaviors among agents as possible, the latter being quantified by a user-input metric that would simply need to be passed on to the RL calibrator agent's reward function. We haven't specifically explored this aspect in the present paper, but we believe that there is significant potential for research attempting to design and learn fair and ethical targets for equilibria using our algorithm in multi-agent systems.

## Acknowledgments and Disclosure of Funding

This work was produced while all authors were employed at JPMorgan Chase & Co, in the Artificial Intelligence Research group. We would like to thank Cathy Lin, Chi Nzelu and Eddie Wen from J.P. Morgan – Corporate and Investment Bank, for their support and inputs that provided the impetus and motivation for this work. We would also like to thank the J.P. Morgan leadership for providing the vision and sponsorship for J.P. Morgan AI Research.

## Disclaimer

This paper was prepared for information purposes by the Artificial Intelligence Research group of JPMorgan Chase & Co and its affiliates ("JP Morgan"), and is not a product of the Research Department of JP Morgan. JP Morgan makes no representation and warranty whatsoever and disclaims all liability, for the completeness, accuracy or reliability of the information contained herein. This document is not intended as investment research or investment advice, or a recommendation, offer or solicitation for the purchase or sale of any security, financial instrument, financial product or service, or to be used in any way for evaluating the merits of participating in any transaction, and shall not constitute a solicitation under any jurisdiction or to any person, if such solicitation under such jurisdiction or to such person would be unlawful.

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
