[Supplementary Material]

# Calibration of Shared Equilibria in General Sum Partially Observable Markov Games - Supplementary

**Nelson Vadori, Sumitra Ganesh, Prashant Reddy, Manuela Veloso**
J.P. Morgan AI Research
{nelson.n.vadori, sumitra.ganesh, prashant.reddy, manuela.veloso}@jpmorgan.com

## A  Proofs

**Lemma 2.** *Every extended transitive game with payoff $f$ has at least one $(f, \epsilon)$-self-play sequence for every $\epsilon > 0$, and every such sequence is finite.*

*Proof.* First note that such a game has at least one $(f, \epsilon)$-self-play sequence for every $\epsilon > 0$ since every $(x, x)$ is a $(f, \epsilon)$-self-play sequence of size 0 (cf. definition 2). Then, let $(x_n, y_n)$ be a $(f, \epsilon)$-self-play sequence. By definition of the self-play sequence we have $f(x_{2n+1}, x_{2n}) > f(x_{2n}, x_{2n}) + \epsilon$. By extended transitivity (cf. assumption 1) this implies $T(x_{2n+1}) > T(x_{2n}) + \delta_\epsilon$. But $x_{2n+1} = x_{2n+2}$ by definition of the self-play sequence, hence $T(x_{2n+2}) > T(x_{2n}) + \delta_\epsilon$. By induction $T(x_{2n}) > T(x_0) + n\delta_\epsilon$ for $n \geq 1$. If the sequence is not finite and since $\delta_\epsilon > 0$, one can take the limit as $n \to \infty$ and get a contradiction, since $T$ is bounded by extended transitivity assumption. □

**Theorem 2.** *An extended transitive game with payoff $f$ has a symmetric pure strategy $\epsilon-$Nash equilibrium for every $\epsilon > 0$, which further can be reached within a finite number of steps following a $(f, \epsilon)$-self-play sequence.*

*Proof.* Let $\epsilon > 0$. Take a $(f, \epsilon)$-self-play sequence. By lemma 2, such a sequence exists and is finite, hence one may take a $(f, \epsilon)$-self-play sequence of maximal size, say $2N_\epsilon$. Assume that its end point $(x, x)$ is not an $\epsilon-$Nash equilibrium. Then $\exists y$: $f(y, x) > f(x, x) + \epsilon$, which means that one can extend the $(f, \epsilon)$-self-play sequence to size $2N_\epsilon + 2$ with entries $(y, x)$ and $(y, y)$, which violates the fact that such a sequence was taken of maximal size. □

**Theorem 3.** *An extended transitive game with continuous payoff $f$ and compact strategy set has a symmetric pure strategy Nash equilibrium.*

*Proof.* By theorem 2, take a sequence of $\epsilon_n$-Nash equilibria with $\epsilon_n \to 0$ and corresponding $(f, \epsilon_n)$-self-play sequence endpoints $(x_n, x_n)$. By compactness assumption, this sequence has a converging subsequence $(x_{m_n}, x_{m_n})$, whose limit point $(x_*, x_*)$ belongs to the strategy set. We have by definition of $\epsilon_{m_n}$-Nash equilibrium that $f(x_{m_n}, x_{m_n}) \geq \sup_y f(y, x_{m_n}) - \epsilon_{m_n}$. Taking the limit as $n \to \infty$ and using continuity of $f$, we get $f(x_*, x_*) \geq \sup_y f(y, x_*)$, which shows that $(x_*, x_*)$ is a symmetric pure strategy Nash equilibrium. □

**Theorem 1**. Let $\mathbf{\Lambda}$ be a supertype profile. Assume that the symmetric 2-player game with pure strategy set $\mathcal{X}$ and payoff $\widehat{V}$ is extended transitive. Then, there exists an $\epsilon-$shared equilibrium for every $\epsilon > 0$, which further can be reached within a finite number of steps following a $(\widehat{V}, \epsilon)$-self-play sequence. Further, if $\mathcal{S}$, $\mathcal{A}$ and $\mathcal{S}^\lambda$ are finite and the rewards $\mathcal{R}$ are bounded, then there exists a shared equilibrium. In particular, if $(\pi_{\theta_n})_{n \geq 0}$ is a sequence of policies obtained following the gradient update (4) with $\widehat{V}(\pi_{\theta_{n+1}}, \pi_{\theta_n}) > \widehat{V}(\pi_{\theta_n}, \pi_{\theta_n}) + \epsilon$, then $(\pi_{\theta_n})_{n \geq 0}$ generates a finite $(\widehat{V}, \epsilon)$-self-play sequence and its endpoint $(\pi_\epsilon, \pi_\epsilon)$ is an $\epsilon-$shared equilibrium.

*Proof.* The first part of the theorem follows from theorem 2. Then, we have by assumption that $\mathcal{S}$, $\mathcal{A}$, $\mathcal{S}^\lambda$ are finite. Denote $m := |\mathcal{S}| \cdot |\mathcal{A}| \cdot |\mathcal{S}^\lambda|$. In that case $\mathcal{X}$ is given by:

$$\mathcal{X} = \{(x_a^{s,\lambda}) \in [0,1]^m : \forall s \in [1,|\mathcal{S}|], \lambda \in [1,|\mathcal{S}^\lambda|], \sum_{a=1}^{|\mathcal{A}|} x_a^{s,\lambda} = 1\}$$

$\mathcal{X}$ is a closed and bounded subset of $[0,1]^m$, hence by Heine–Borel theorem it is compact. Note that closedness comes from the fact that summation to 1 is preserved by passing to the limit. By assumption, the rewards are bounded, so by lemma 1, $V_{\Lambda_i}$ is continuous for all $i$, which yields continuity of $\widehat{V}$, hence we can apply theorem 3 to conclude.

Finally, if $(\pi_{\theta_n})_{n \geq 0}$ is a sequence of policies obtained following the gradient update (4) with $\widehat{V}(\pi_{\theta_{n+1}}, \pi_{\theta_n}) > \widehat{V}(\pi_{\theta_n}, \pi_{\theta_n}) + \epsilon$, then the self-play sequence generated by $(\pi_{\theta_n})_{n \geq 0}$ is finite by lemma 2, and its endpoint is necessarily a symmetric pure strategy $\epsilon$-Nash equilibrium according to the proof of theorem 2, hence an $\epsilon$-shared equilibrium. $\qquad\square$

**Lemma 1.** Assume that the rewards $\mathcal{R}$ are bounded, and that $\mathcal{S}$, $\mathcal{A}$ and $\mathcal{S}^\lambda$ are finite. Then $V_{\Lambda_i}$ is continuous on $\mathcal{X} \times \mathcal{X}$ for all $i$, where $\mathcal{X}$ is equipped with the total variation metric.

*Proof.* Since by assumption $\mathcal{S}$, $\mathcal{A}$ and $\mathcal{S}^\lambda$ are finite, we will use interchangeably sum and integral over these spaces. Let us denote the total variation metric for probability measures $\pi_1, \pi_2$ on $\mathcal{X}$:

$$\rho_{TV}(\pi_1, \pi_2) := \frac{1}{2} \max_{s,\lambda} \sum_{a \in \mathcal{A}} |\pi_1(a|s,\lambda) - \pi_2(a|s,\lambda)|$$

and let us equip the product space $\mathcal{X} \times \mathcal{X}$ with the metric:

$$\rho_{TV}((\pi_1, \pi_2), (\pi_3, \pi_4)) := \rho_{TV}(\pi_1, \pi_3) + \rho_{TV}(\pi_2, \pi_4).$$

Remember that $z_t^{(i)} := (s_t^{(i)}, a_t^{(i)}, \lambda_i)$. Let:

$$V_{\Lambda_i}(\pi_1, \pi_2, \boldsymbol{s}, \boldsymbol{\lambda}) := \mathbb{E}_{a_t^{(i)} \sim \pi_1(\cdot|\cdot,\lambda_i), \, a_t^{(j)} \sim \pi_2(\cdot|\cdot,\lambda_j)} \left[ \sum_{t=0}^{\infty} \gamma^t \mathcal{R}(z_t^{(i)}, \boldsymbol{z_t^{(-i)}}) | s_0 = s \right], \quad j \neq i,$$

so that:

$$V_{\Lambda_i}(\pi_1, \pi_2) = \int_{\boldsymbol{s}} \int_{\boldsymbol{\lambda}} V_{\Lambda_i}(\pi_1, \pi_2, \boldsymbol{s}, \boldsymbol{\lambda}) \cdot \Pi_{j=1}^{n}[\mu_{\lambda_j}^0(ds_j) p_{\Lambda_j}(d\lambda_j)]$$

Then we have:

$$V_{\Lambda_i}(\pi_1, \pi_2, \boldsymbol{s}, \boldsymbol{\lambda}) = \int_{\boldsymbol{a}} \mathcal{R}(z^{(i)}, \boldsymbol{z^{(-i)}}) \pi_1(da_i|s_i, \lambda_i) \Pi_{j \neq i} \pi_2(da_j|s_j, \lambda_j)$$

$$+ \gamma \int_{\boldsymbol{a}} \int_{\boldsymbol{s'}} \mathcal{T}(\boldsymbol{z}, d\boldsymbol{s'}) V_{\Lambda_i}(\pi_1, \pi_2)(\boldsymbol{s'}, \boldsymbol{\lambda}) \pi_1(da_i|s_i, \lambda_i) \Pi_{j \neq i} \pi_2(da_j|s_j, \lambda_j)$$

The goal is to compute $|V_{\Lambda_i}(\pi_1, \pi_2, \boldsymbol{s}, \boldsymbol{\lambda}) - V_{\Lambda_i}(\pi_3, \pi_4, \boldsymbol{s}, \boldsymbol{\lambda})|$ and show that the latter is small provided that $\rho_{TV}((\pi_1, \pi_2), (\pi_3, \pi_4))$ is small. Let us use the notation:

$$c_1(\pi_1, \pi_2) := \int_{\boldsymbol{a}} \mathcal{R}(z^{(i)}, \boldsymbol{z^{(-i)}}) \pi_1(da_i|s_i, \lambda_i) \Pi_{j \neq i} \pi_2(da_j|s_j, \lambda_j)$$

Since by assumption $|\mathcal{R}|$ is bounded, say by $\mathcal{R}_{max}$ we have:

$|c_1(\pi_1, \pi_2) - c_1(\pi_3, \pi_4)|$

$\leq \mathcal{R}_{max} \int_{\boldsymbol{a}} |\pi_1(da_i|s_i, \lambda_i) \Pi_{j \neq i} \pi_2(da_j|s_j, \lambda_j) - \pi_3(da_i|s_i, \lambda_i) \Pi_{j \neq i} \pi_4(da_j|s_j, \lambda_j)|$

$\leq \mathcal{R}_{max} \int_{a_i} |\pi_1(da_i|s_i, \lambda_i) - \pi_3(da_i|s_i, \lambda_i)|$

$+ \mathcal{R}_{max} \int_{\boldsymbol{a_{-i}}} |\Pi_{j \neq i} \pi_2(da_j|s_j, \lambda_j) - \Pi_{j \neq i} \pi_4(da_j|s_j, \lambda_j)|$

$\leq 2\mathcal{R}_{max} \rho_{TV}(\pi_1, \pi_3) + 2\mathcal{R}_{max}(n-1)\rho_{TV}(\pi_2, \pi_4) \leq 2n\mathcal{R}_{max}\rho_{TV}((\pi_1, \pi_2), (\pi_3, \pi_4))$

Now, let us use the notation:

$$c_2(\pi_1, \pi_2) := \int_{\boldsymbol{a}} \int_{\boldsymbol{s'}} \mathcal{T}(\boldsymbol{z}, d\boldsymbol{s'}) V_{\Lambda_i}(\pi_1, \pi_2)(\boldsymbol{s'}, \boldsymbol{\lambda}) \pi_1(da_i|s_i, \lambda_i) \Pi_{j \neq i} \pi_2(da_j|s_j, \lambda_j)$$

For the term $|c_2(\pi_1, \pi_2) - c_2(\pi_3, \pi_4)|$, we can split:

$$V_{\Lambda_i}(\pi_1, \pi_2)(\boldsymbol{s'}, \boldsymbol{\lambda}) \pi_1(da_i|s_i, \lambda_i) \Pi_{j \neq i} \pi_2(da_j|s_j, \lambda_j)$$
$$- V_{\Lambda_i}(\pi_3, \pi_4)(\boldsymbol{s'}, \boldsymbol{\lambda}) \pi_3(da_i|s_i, \lambda_i) \Pi_{j \neq i} \pi_4(da_j|s_j, \lambda_j)$$
$$= V_{\Lambda_i}(\pi_1, \pi_2)(\boldsymbol{s'}, \boldsymbol{\lambda})[\pi_1(da_i|s_i, \lambda_i)\Pi_{j \neq i}\pi_2(da_j|s_j, \lambda_j) - \pi_3(da_i|s_i, \lambda_i)\Pi_{j \neq i}\pi_4(da_j|s_j, \lambda_j)]$$
$$+ \pi_3(da_i|s_i, \lambda_i)\Pi_{j \neq i}\pi_4(da_j|s_j, \lambda_j)[V_{\Lambda_i}(\pi_1, \pi_2)(\boldsymbol{s'}, \boldsymbol{\lambda}) - V_{\Lambda_i}(\pi_3, \pi_4)(\boldsymbol{s'}, \boldsymbol{\lambda})]$$

Since $V_{\Lambda_i}$ is bounded by $\mathcal{R}_{max}(1 - \gamma)^{-1}$, and noting that we have, as for $c_1$, that:

$$\int_{\boldsymbol{a}} |\pi_1(da_i|s_i, \lambda_i)\Pi_{j \neq i}\pi_2(da_j|s_j, \lambda_j) - \pi_3(da_i|s_i, \lambda_i)\Pi_{j \neq i}\pi_4(da_j|s_j, \lambda_j)|$$
$$\leq 2n\rho_{TV}((\pi_1, \pi_2), (\pi_3, \pi_4))$$

we then have:

$$|c_2(\pi_1, \pi_2) - c_2(\pi_3, \pi_4)| \leq 2n\mathcal{R}_{max}(1 - \gamma)^{-1}\rho_{TV}((\pi_1, \pi_2), (\pi_3, \pi_4))$$
$$+ \max_{\boldsymbol{s}, \boldsymbol{\lambda}} |V_{\Lambda_i}(\pi_1, \pi_2, \boldsymbol{s}, \boldsymbol{\lambda}) - V_{\Lambda_i}(\pi_3, \pi_4, \boldsymbol{s}, \boldsymbol{\lambda})|$$

We then have, collecting all terms together:

$$|V_{\Lambda_i}(\pi_1, \pi_2, \boldsymbol{s}, \boldsymbol{\lambda}) - V_{\Lambda_i}(\pi_3, \pi_4, \boldsymbol{s}, \boldsymbol{\lambda})| \leq 2n\mathcal{R}_{max}(1 + \gamma(1 - \gamma)^{-1})\rho_{TV}((\pi_1, \pi_2), (\pi_3, \pi_4))$$
$$+ \gamma \max_{\boldsymbol{s}, \boldsymbol{\lambda}} |V_{\Lambda_i}(\pi_1, \pi_2, \boldsymbol{s}, \boldsymbol{\lambda}) - V_{\Lambda_i}(\pi_3, \pi_4, \boldsymbol{s}, \boldsymbol{\lambda})|$$

Taking the maximum over $\boldsymbol{s}, \boldsymbol{\lambda}$ on the left hand-side and rearranging terms finally yields:

$$|V_{\Lambda_i}(\pi_1, \pi_2) - V_{\Lambda_i}(\pi_3, \pi_4)| \leq \max_{\boldsymbol{s}, \boldsymbol{\lambda}} |V_{\Lambda_i}(\pi_1, \pi_2, \boldsymbol{s}, \boldsymbol{\lambda}) - V_{\Lambda_i}(\pi_3, \pi_4, \boldsymbol{s}, \boldsymbol{\lambda})|$$

$$\leq 2n(1 - \gamma)^{-1}\mathcal{R}_{max}(1 + \gamma(1 - \gamma)^{-1})\rho_{TV}((\pi_1, \pi_2), (\pi_3, \pi_4))$$
$$= 2n(1 - \gamma)^{-2}\mathcal{R}_{max}\rho_{TV}((\pi_1, \pi_2), (\pi_3, \pi_4))$$

which yields the desired continuity result. $\square$

# B Experiments: details and complete set of results

## B.1 Description of the n-player market setting and of the merchant agents on a shared policy

We implemented a simulator of a market where **merchant** agents $i$ offer prices $\boldsymbol{p}^{(i)}_{buy,t}$, $\boldsymbol{p}^{(i)}_{sell,t}$ to **customers** at which they are willing to *buy* and *sell* a certain good, for example coffee, from/to them. A given merchant $i$ cannot observe the prices that his competitors $j \neq i$ are offering to customers, hence the partially observed setting.

There exists a **reference facility** that all merchants and customers can observe and can transact with at buy/sell prices $\boldsymbol{p}^*_{buy,t}$, $\boldsymbol{p}^*_{sell,t}$ publicly available at all times. Consequently, if a merchant offers to a customer a price less attractive than the reference price, the customer will prefer to transact with the reference facility instead. $\boldsymbol{p}^*_{buy,t}$, $\boldsymbol{p}^*_{sell,t}$ are assumed to be of the form $\boldsymbol{p}^*_{buy,t} = m^*_t - \boldsymbol{\delta}^*_t$, $\boldsymbol{p}^*_{sell,t} = m^*_t + \boldsymbol{\delta}^*_t$, where both $m^*_t, \boldsymbol{\delta}^*_t$ have Gaussian increments over each timestep and $\boldsymbol{\delta}^*_t \geq 0$.

A **merchant $i$'s inventory** $q^{(i)}_t$ is the net quantity of good that remains in his hands as a result of all transactions performed with customers and the reference facility up to time $t$. We assume that it is permitted to sell on credit, so that inventory $q^{(i)}_t$ can be negative.

We have denoted the prices in bold letters since these prices are in fact functions of the quantity $q$ that is being transacted, for example $\boldsymbol{p}^{(i)}_{buy,t} \equiv q \to p^{(i)}_{buy,t}(q)$. In order to simplify the setting, we

assume that **merchants' actions** $a_t^{(i)}$ only consist in i) specifying multiplicative buy/sell factors $\epsilon_{t,b}^{(i)}$, $\epsilon_{t,s}^{(i)} \in [-1,1]$ on top of the reference curve to generate price curves: $\boldsymbol{p}_{buy,t}^{(i)} := \boldsymbol{p}_{buy,t}^*(1 + \epsilon_{t,b}^{(i)})$, $\boldsymbol{p}_{sell,t}^{(i)} := \boldsymbol{p}_{sell,t}^*(1 + \epsilon_{t,s}^{(i)})$ and ii) specifying a fraction $h_t^{(i)} \in [0,1]$ of current inventory $q_t^{(i)}$ to transact at the reference facility, so that $a_t^{(i)} = (\epsilon_{t,b}^{(i)}, \epsilon_{t,s}^{(i)}, h_t^{(i)}) \in [-1,1]^2 \times [0,1]$. The **merchant's state** $s_t^{(i)} \in \mathbb{R}^d$ with $d \sim 500$ consist in the reference price and his recent transaction history with all customers, in particular his inventory.

**Merchants' rewards** depend on other merchant's prices and consist in the profit made as a result of i) transactions performed with customers and the reference facility and ii) the change in inventory's value due to possible fluctuations of the reference price.

**Customers** are assumed at every point in time $t$ to either want to buy or sell with equal probability a certain quantity. We split 500 customers into 10 **customer clusters**, cluster $i \in [1,10]$ being associated to quantity $i$. For example, a customer belonging to cluster 5 will generate transactions of quantity 5.

**Types and supertypes.** In our setting, merchants differ by 1) their **connectivity** to customers (they can transact only with connected customers) and 2) their **inventory tolerance** factor $\xi_i$, which penalizes holding a large inventory by adding a term $-\xi_i|q_t^{(i)}|$ to their reward. We define the **supertype** $\Lambda_i$ **as a vector of size 12**: 10 probabilities of being connected to customers belonging to the 10 customer clusters, plus the mean and standard deviation of the normal distribution generating the merchant's inventory tolerance coefficient $\xi_i$ [1]. In a given episode, a merchant may be connected differently to customers in the same cluster, however he has the same probability to be connected to them. That means that the **type** $\lambda_i$ sampled probabilistically at the beginning of each episode is a vector of size 11: 10 entries in $[0,1]$ corresponding to the sampled fractions of connected customers in each one of the 10 clusters, and 1 inventory tolerance factor. For example, if a merchant has in its supertype a probability 30% to be connected to customers in cluster 5, then each one of the 50 binary connections between the merchant and customers of cluster 5 will be sampled independently at the beginning of the episode as a Bernoulli random number with associated probability 30%, and the resulting fraction of connected customers is recorded in $\lambda_i$.

**Calibration targets.** We consider calibration targets of two different types. The **market share** of a specific merchant $i$ is defined as the fraction of the sum of all customers' transaction quantities (over an episode) that merchant $i$ has obtained. Note that the sum of market share over merchant's doesn't necessarily sum to 1 since customers can transact with the reference facility if the merchant's prices are not attractive enough. The **transaction distribution** is defined as percentiles of the distribution - over an episode - of transaction quantities per timestep received by a merchant as a result of his interactions with all customers.

## B.2   RL calibrator agent

As described in section 3, the state of the RL calibrator is the current supertype profile $\boldsymbol{\Lambda}$ and its action is a supertype profile increment $\boldsymbol{\delta\Lambda}$. In section B.1, we described the supertype $\Lambda_i$ for each merchant as a vector of size 12, and in section 4 we mentioned that we conducted experiments using 2 distinct supertypes for the 5-10 merchant agents (see also section B.3 for a more detailed description). As a result, both the calibrator's state $\boldsymbol{\Lambda}$ and action $\boldsymbol{\delta\Lambda}$ consist of the 12 supertype entries for 2 distinct supertypes, i.e. 24 real numbers. In our experiments, we set the standard deviation of the normal distribution associated to inventory tolerance of supertype 1 to be zero since supertype 1 is associated to 1 merchant only, which reduces the size to 23 real numbers. The corresponding ranges for the parameters $(\Lambda_i(j))_{j=1..12}$ in the RL calibrator's policy action and state spaces are reported in table 1.

Table 1: RL calibrator state and action spaces.

| Supertype parameter flavor $j$ | state $\Lambda_i(j)$ range | action $\delta\Lambda_i(j)$ range |
|---|---|---|
| customer cluster connectivity probability | $[0, 1]$ | $[-1, 1]$ |
| inventory tolerance Gaussian mean | $[0, 5]$ | $[-5, 5]$ |
| inventory tolerance Gaussian stDev | $[0, 2]$ | $[-2, 2]$ |

As mentioned in section 3, the calibrator agent's reward $r_b^{cal}$ associated to an episode $b$ is given by

$$r_b^{cal} = \sum_{k=1}^{K} w_k \overbrace{\ell_k^{-1}(f_*^{(k)} - f_{cal}^{(k)}((\boldsymbol{z_{t,b}})_{t \geq 0}))}^{r_b^{(k)}} \qquad (1)$$

We give in table 3 a breakdown of these sub-objectives for each experiment (for each experiment, all $K$ sub-objectives are required to be achieved simultaneously, cf. equation (1) above). Table 3 is associated with reward functions mentioned below, where we denote $m_{super_1} = m_{super_1}((\boldsymbol{z_t})_{t \geq 0})$ the market share of supertype 1 observed throughout an episode, $m_{total}$ the sum of all merchants' marketshares, $\widehat{v}_{super_j}(p)$ the observed $(10p)^{th}\%$ percentile of supertype $j$'s transaction distribution per timestep.

In experiment 1, $r = (1 + r^{(1)} + 0.2r^{(2)})^{-1}$, with $v_{super_1} = [8, 8, 8, 9, 9, 9, 10, 10, 10]$, $r^{(1)} = \frac{1}{2}(\max(0.15 - m_{super_1}, 0) + \max(0.8 - m_{total}, 0))$, $r^{(2)} = \frac{1}{9}\sum_{p=1}^{9}|v_{super_1}(p) - \widehat{v}_{super_1}(p)|$.

In experiment 2/3, $r = (1 + r^{(1)} + 0.2r^{(2)} + 0.2r^{(3)})^{-1}$, with $v_{super_1} = [8, 8, 8, 9, 9, 9, 10, 10, 10]$, $v_{super_2} = [2, 3, 3, 4, 5, 5, 6, 6, 7]$, $r^{(1)} = \frac{1}{2}(\max(0.15 - m_{super_1}, 0) + \max(0.8 - m_{total}, 0))$, $r^{(j+1)} = \frac{1}{9}\sum_{p=1}^{9}|v_{super_j}(p) - \widehat{v}_{super_j}(p)|, j \in \{1, 2\}$.

In experiment 4, $r = (1 + r^{(1)} + r^{(2)})^{-1}$, with $r^{(1)} = |0.25 - m_{super_1}|$, $r^{(2)} = \max(0.8 - m_{total}, 0)$.
In experiment 5, $r = (1 + r^{(1)} + r^{(2)})^{-1}$, with $r^{(1)} = |0.4 - m_{super_1}|$, $r^{(2)} = |0.8 - m_{total}|$.

### B.3   Details on experiments

Experiments were conducted in the RLlib multi-agent framework [1], ran on AWS using a EC2 C5 24xlarge instance with 96 CPUs, resulting in a training time of approximately 1 day per experiment.

The 5 experiments we conducted are described in table 2, with a calibration target breakdown in table 3 (see section B.1 for a description of the market setting and merchant agents, and section B.2 for a description of the RL calibrator agent's state, actions and rewards). For example, according to table 3, in experiment 1, we calibrate 23 parameters altogether in order to achieve 11 calibration targets simultaneously. As mentioned in section 4, in all experiments, merchant 1 was assigned supertype 1, and all $n - 1$ other merchants were assigned supertype 2.

**Shared Policy and calibrator's policy**. Both policies were trained jointly according to algorithm 1 using Proximal Policy Optimization [3], an extension of TRPO [2]. We used configuration parameters in line with [3], that is a clip parameter of 0.3, an adaptive KL penalty with a KL target of 0.01 (so as to smoothly vary the supertype profile) and a learning rate of $10^{-4}$. We found that entropy regularization was not specifically helpful in our case. Episodes were taken of length 60 time steps with a discount factor of 1, using $B = 90$ parallel runs in between policy updates (for both policies). As a result, each policy update was performed with a batch size of $n \cdot 60 \cdot 90$ timesteps for the shared policy, and $3 \cdot 90$ timesteps for the calibrator's policy, as we allowed the calibrator to take 3 actions per episode (that is, updating the supertype profile $\boldsymbol{\Lambda}$ 3 times), together with 30 iterations of stochastic gradient descent. We used for each policy a fully connected neural net with 2 hidden layers, 256 nodes per layer, and tanh activation. Since our action space is continuous, the outputs of the neural net are the mean and stDev of a standard normal distribution, which is then used to sample actions probabilistically (the covariance matrix across actions is chosen to be diagonal).

**Bayesian optimization baseline**. We used Bayesian optimization to suggest a next supertype profile $\boldsymbol{\Lambda}$ to try next, every $M$ training iterations of the shared policy. That is, every $M$ training iterations, we

record the calibrator's reward as in section B.2, and use Bayesian optimization to suggest the next best $\Lambda$ to try. We empirically noticed that if $M$ was taken too low ($M \sim 10$), the shared policy couldn't adapt as the supertype profile changes were too frequent (and potentially too drastic), thus leading to degenerate behaviors (e.g. merchants not transacting at all). We tested values of $M = 10$, $M = 50$, $M = 100$, $M = 200$, and opted for $M = 100$ as we found it was a good trade-off between doing sufficiently frequent supertype profile updates and at the same time giving enough time to the shared policy to adapt. We chose an acquisition function of upper confidence bound (UCB) type [4]. Given the nature of our problem where agents on the shared policy need to be given sufficient time to adapt to a new supertype profile choice $\Lambda$, we opted for a relatively low UCB exploration parameter of $\kappa = 0.5$, which we empirically found yielded a good trade-off between exploration and exploitation (taking high exploration coefficient can yield drastic changes in the supertype profile space, which can prevent agents to learn correctly an equilibrium). In figure 1 we look - in the case of experiment 1 - at the impact of the choice of $M$ in the EI (expected improvement) and UCB (exploration parameter of $\kappa = 1.5$) cases and find that different choices of $M$ and of the acquisition function yield similar performance. We also look at the case "CALSHEQ_no_state" where the calibrator policy directly samples supertype values (rather than increments) without any state information (i.e. the calibrator policy's action is conditioned on a constant), and find that it translates into a significant decrease in performance. We further note that decreasing $M$ has a cost, especially when $\Lambda$ is high dimensional, since the BO step will become more and more expensive with increasing observation history length. For example, in the case of experiment 1, we observed with $M = 1$ that the training hadn't reached the 20M timestep budget after 2 days (for a calibrator reward in line with other values of $M$). The covariance function of the Gaussian process was set to a Matern kernel with $\nu = 2.5$.

Figure 1: Calibrator Reward during training for various number of BO frequencies $M$ - Experiment 1 - **(Left)** BO Expected Improvement (EI) - **(Right)** BO UCB with exploration parameter $\kappa = 1.5$.

Table 2: Summary of experiment configuration.

| Experiment # | # Merchant Agents | Budget # Training Steps ($10^6$) | # distinct Supertypes | # Supertype parameters to be calibrated | Total # Calibration Targets |
|---|---|---|---|---|---|
| 1 | 5 | 40 | 2 | 20 | 11 |
| 2 | 5 | 40 | 2 | 20 | 20 |
| 3 | 10 | 20 | 2 | 20 | 20 |
| 4 | 10 | 20 | 2 | 23 | 2 |
| 5 | 5 | 40 | 2 | 23 | 2 |

## B.4 Complete set of experimental results associated to section 4

In this section we display the complete set of results associated to figures shown in section 4. We display in figure 2 the rewards of all agents during training (calibrator, merchant on supertype 1 and $n - 1$ merchants on supertype 2) for experiments 1-5 previously described. In figures 3-7 we display the calibration fits for all calibration targets described in table 3 (we reiterate that for each experiment,

Table 3: Calibration target breakdown

| Experiment # | # Calibration Targets | Calibration Target Type |
|:---:|:---:|:---:|
| 1 | 9 | transaction quantity distribution supertype 1<br>percentiles $10\% - 90\%$ target 8, 8, 8, 9, 9, 9, 10, 10, 10 |
|  | 2 | market share supertype $1 \geq 15\%$ + total $\geq 80\%$ |
| 2 | 18 | transaction quantity distribution Supertypes 1+2<br>supertype 1 - percentiles $10\% - 90\%$ target 8, 8, 8, 9, 9, 9, 10, 10, 10<br>supertype 2 - percentiles $10\% - 90\%$ target 2, 3, 3, 4, 5, 5, 6, 6, 7 |
|  | 2 | market share supertype $1 \geq 15\%$ + total $\geq 80\%$ |
| 3 | 18 | transaction quantity distribution Supertypes 1+2<br>supertype 1 - percentiles $10\% - 90\%$ target 8, 8, 8, 9, 9, 9, 10, 10, 10<br>supertype 2 - percentiles $10\% - 90\%$ target 2, 3, 3, 4, 5, 5, 6, 6, 7 |
|  | 2 | market share supertype $1 \geq 15\%$ + total $\geq 80\%$ |
| 4 | 1 | market share supertype $1 = 25\%$ |
|  | 1 | total market share $\geq 80\%$ |
| 5 | 1 | market share supertype $1 = 40\%$ |
|  | 1 | total market share $= 80\%$ |

all calibration targets are required to be achieved simultaneously). In figures 8-12 we display the calibrated parameters associated to the calibration fits, that is the parameters in supertypes 1 and 2 (customer connectivity probability and inventory tolerance Gaussian mean and stDev) that allow to reach the calibration targets. As discussed in section 4, *CALSHEQ* outperforms BO in terms of efficiency, accuracy of the fit, and smoothness. The shaded areas correspond to 1 stDev, computed according to the so-called range rule $\frac{(\max - \min)}{4}$. In the case of *CALSHEQ*, we plot the mean of the calibration targets and calibrated parameters over the $B$ episodes.

Figure 2: Rewards during training, averaged over episodes $B$ - Calibrator **(Top)**, Supertypes 1/2 **(Mid/Bottom)** - experiments 1-2-3-4-5, respectively 5-5-10-10-5 agents. *CALSHEQ* (ours) and baseline (Bayesian optimization). Shaded area represents $\pm 1$ stDev.

Figure 3: Experiment 1 - Calibration target fit for transaction quantity distribution percentile and Market Share during training, averaged over episodes $B$. Dashed line target indicates that the constraint was set to be greater than target (not equal to it). *CALSHEQ* (ours) and baseline (Bayesian optimization).

Figure 4: Experiment 2 - Calibration target fit for transaction quantity distribution percentile and Market Share during training, averaged over episodes $B$. Dashed line target indicates that the constraint was set to be greater than target (not equal to it). *CALSHEQ* (ours) and baseline (Bayesian optimization).

Figure 5: Experiment 3 - Calibration target fit for transaction quantity distribution percentile and Market Share during training, averaged over episodes $B$. Dashed line target indicates that the constraint was set to be greater than target (not equal to it). *CALSHEQ* (ours) and baseline (Bayesian optimization).

Figure 6: Experiment 4 - Calibration target fit for Market Share during training, averaged over episodes $B$. Dashed line target indicates that the constraint was set to be greater than target (not equal to it). *CALSHEQ* (ours) and baseline (Bayesian optimization).

Figure 7: Experiment 5 - Calibration target fit for Market Share during training, averaged over episodes $B$. *CALSHEQ* (ours) and baseline (Bayesian optimization).

Figure 8: Experiment 1 - Calibrated parameters, averaged over episodes $B$. *CALSHEQ* (ours) and baseline (Bayesian optimization).

Figure 9: Experiment 2 - Calibrated parameters, averaged over episodes $B$. *CALSHEQ* (ours) and baseline (Bayesian optimization).

Figure 10: Experiment 3 - Calibrated parameters, averaged over episodes $B$. *CALSHEQ* (ours) and baseline (Bayesian optimization).

Figure 11: Experiment 4 - Calibrated parameters, averaged over episodes $B$. *CALSHEQ* (ours) and baseline (Bayesian optimization).

Figure 12: Experiment 5 - Calibrated parameters, averaged over episodes $B$. *CALSHEQ* (ours) and baseline (Bayesian optimization).

## Disclaimer

This paper was prepared for information purposes by the Artificial Intelligence Research group of JPMorgan Chase & Co and its affiliates ("JP Morgan"), and is not a product of the Research Department of JP Morgan. JP Morgan makes no representation and warranty whatsoever and disclaims all liability, for the completeness, accuracy or reliability of the information contained herein. This document is not intended as investment research or investment advice, or a recommendation, offer or solicitation for the purchase or sale of any security, financial instrument, financial product or service, or to be used in any way for evaluating the merits of participating in any transaction, and shall not constitute a solicitation under any jurisdiction or to any person, if such solicitation under such jurisdiction or to such person would be unlawful.

## Footnotes

[1] in experiments of section B.3, we set the standard deviation of the normal distribution associated to inventory tolerance of supertype 1 to be zero since supertype 1 is associated to 1 merchant only