[Reviews · NeurIPS 2020]

Review 1

Summary and Contributions: Overall I very much appreciated the rebuttal and the definition of the symmetric 2 player game seems to make more sense now. Also, the response on existence of self-play sequence seems convincing. I do still think that this paper would be much stronger if it could more clearly describe what games this really applies to. ----- Motivated by the success of parameter sharing in multiagent RL, this paper explores symmetric games, and their equilibria. Specifically, it presents a results that self-play can converge to a 'shared equilibrium' (a pure strategy of the symmetric game). In the second part of the paper, the issue of 'calibration' is investigated. The idea is that if one uses MARL as the basis for simulation (agent-based modeling), one wants to calibrate the simulation such that the resulting behavior agrees with the reality. For this, a new learning algorithm is introduced.

Strengths: I like the idea of further formalizing and understanding the otherwise somewhat vague notion of "self-play". Also the idea of having a learning algorithm that can directly calibrate for ABM applications is interesting. Overall the paper is written in a clear and easy to follow style.

Weaknesses: It is a bit unclear why the first (shared equilibria) and second (calibration) parts of this paper are really put together. Why are these not two separate papers? The results do not seem to depend on each other. Some of the technical points are not completely clear. There could be mistakes. The assumption of 'extended transitivity' is extremely strong. Essentially it seems to directly implies the existence of an optimum payoff in symmetric policies. Given such strong assumptions, theorem 1 seems not so surprising. The relation to literature could be strengthened. For instance, an important question is how the identified class of games here relates to potential games, for which we know that better reply dynamics converge to a Nash equilibrium? What is the relation to Bayesian games (that also have types)? Results seem OK, but in most cases the proposed method does not seem to do much better than the baseline (Bayesian optimization)

Correctness: The paper critically hinges around the notion of the symmetric game constructed by making use of \hat{V} in (4). It is not clear to me, however, that this is actually a symmetric payoff function. The problem is that a number of the arguments is being hidden. See elaboration below under clarity. Proof of the theorem 1 depends on that of theorem 2 in the appendix. This in turn assumes the existence of a (f,\epsilon)-self play sequence, but it it not clear to me that one can assume this?

Clarity: Overall, the style of writing is clear. However, some of the formulas are not so clear. In particular, from line 126 onwards, the paper depends on a transformation from the n-player game to a 2 player symmetric game. It is not clear to me, however, that this actually is a symmetric game. In particular, (2) defines (in my notation) V_i (\pi_i, \pi_{-i}; \Lambda_i, \Lambda_{-i} ) the payoff to player i, when -it uses \pi_i and has type distribution \Lambda_i, -all other players use \pi_{-i} -all players j!=i have type distribution \Lambda_j for this to define a symmetric game, we need to have that V_i (\pi_i, \pi_{-i}; \Lambda_i, \Lambda_{-i} ) = V_{-i} (\pi_{-i}, \pi_i; \Lambda_i, \Lambda_{-i} ) but I don't think that would hold for arbitrary type distributions per player (those also would need to be permuted appropriately)? Algorithm 1 outputs an 'optimal' calibrator and shared policies. How is optimal defined here? It is not quite clear what the goal of the empirical evaluation really is. What are the columns of figure 1? What does the shading indicate? How many runs is this averaged over? line 251 "π Λ is then updated based on (5) using a classical single-agent RL gradient update." Exactly what type of optimization is used here? Reinforce?

Relation to Prior Work: As mentioned above, I think that the relation to related work could be improved. Particularly, the relation to potential games and Bayesian games seem important omissions. Additionally, there seems to be a clear link to the concepts of "roles" and "anonymity", e.g.: Nair, Ranjit, Tambe, Milind, and Marsella, Stacy, "Team Formation for Reformation", in Proc. of the AAAI Spring Symposium on Intelligent Distributed and Embedded Systems (2002). Wooldridge, Michael, Nicholas R. Jennings, and David Kinny. "A methodology for agent-oriented analysis and design." Proceedings of the third annual conference on Autonomous Agents. 1999. Varakantham, Pradeep, Adulyasak, Yossiri, and Jaillet, Patrick, "Decentralized Stochastic Planning with Anonymity in Interactions", in Proceedings of the Twenty-Eighth AAAI Conference on Artificial Intelligence (2014), pp. 2505--2512. Robbel, Philipp, Oliehoek, Frans A., and Kochenderfer, Mykel J., "Exploiting Anonymity in Approximate Linear Programming: Scaling to Large Multiagent MDPs", in Proceedings of the Thirtieth AAAI Conference on Artificial Intelligence (AAAI) (2016), pp. 2537--2543. Subramanian, Jayakumar and Mahajan, Aditya, "Reinforcement Learning in Stationary Mean-field Games", in Proceedings of the Eighteenth International Conference on Autonomous Agents and Multiagent Systems (AAMAS) (International Foundation for Autonomous Agents and Multiagent Systems, 2019), pp. 251--259.

Reproducibility: No

Additional Feedback:


Review 2

Summary and Contributions: For a known environment, this paper considers the problem of "calibrating" the equilibrium of a multiagent system by adjusting the probabilities of agent types such that the equilibrium of the resulting Bayesian game satisfies given constraints (such as matching action distributions in observed data). The paper focuses on equilibria that are symmetric in the sense that all players share the same policy network; this is without loss of generality, since agent heterogeneity can be entirely subsumed by agent types for a sufficiently rich typespace. After analytically demonstrating that games which satisfy a property called "extended transitivity" are guaranteed to converge to a "shared equilibrium" in self-play, the authors propose a two-timescale algorithm for the calibration problem: the outer loop adjusts the "supertype" profile in an attempt to cause the induced equilibrium to better match the calibration constraints, and the inner loop finds the induced equilibrium. This prevents wasteful re-solving from scratch, because the outer loop changes the supertype profile gradually, unlike Bayesian optimization techniques where there might be large, even discontinuous jumps in supertype values. The performance of the proposed algorithm is compared to Bayesian optimization baselines empirically.

Strengths: The claims are clearly demonstrated and evaluated appropriately. The problem is intriguing and potentially quite significant; this is a to-my-knowledge novel to disciplining multiagent modeling with data. This work is likely to be relevant to the growing contingent of multiagent RL researchers in the NeurIPS community. I especially liked the way in which a game among n agents was shown to be treatable as a two-player game when the policy is shared. This seems like a trick that would be broadly useful for reasoning about multiagent scenarios.

Weaknesses: The extended transitivity assumption is absolutely critical to the results of this paper, but it is not an innocuous assumption. I would have liked to see more discussion of how restrictive this assumption is. Do most scenarios I care about satisfy this property, or is it a pretty special subclass of games?

Correctness: The empirical methodology seems fine; the choice of baselines is reasonable. The analytical claims are very well presented; even when the proof is omitted, it is often clear why the claim is true.

Clarity: The paper is very clearly written. The ideas are built up in a logical order and presented well. I have some minor nitpicks below. - L.44: "Vs." shouldn't be capitalized - Are the types assumed to be sampled independently? - L.219: Assumption 2 is strangely located; should it not be further along? - L.232: "The RL calibrator's state is the (stacked) current values of the supertypes \Lambda_i, ...": Is this different from the current value of the supertype profile? If not, why the new "stacking" terminology?

Relation to Prior Work: Related work is thoroughly surveyed, and the distinctions with the current work are clearly described.

Reproducibility: Yes

Additional Feedback: Overall, this paper clearly describes an effective and interesting solution to a well-motivated problem. I recommend that it be accepted. I do have some questions for the authors: 1. I am a little confused about why we need to introduce the terminology "shared equilibrium". In the Bayesian game formulation, strategies are always a function of types, so is this not just a symmetric Bayes-Nash equilibrium? 2. The conceptual problem being solved by the (outer) calibrator agent is a supervised learning problem rather than an RL problem, so what is the benefit of formalizing the outer optimization as RL? 3. Can you comment on how restrictive or not is the extended transitivity assumption? Should I expect this to hold broadly, or is it a pretty special subclass of games? == post rebuttal == I remain positive about this paper; thanks for the informative rebuttal.


Review 3

Summary and Contributions: In this paper, the authors introduce the concept of shared equilibrium as a symmetric pure Nash equilibrium of a certain class of games and prove the convergence for a certain class of games using self-play. They also propose a dual-Reinforcement Learning approach to match some external targets.

Strengths: This paper focuses on the reachability issues of equilibrium, which is a very important question in MARL and game theory. More specifically, the authors answer the following question: what is the nature of potential equilibria learnt by agents using a shared policy? They showed that the reached equilibria are symmetric pure Nash equilibria of a higher level game on the set of stochastic policies.

Weaknesses: However, the assumptions seem to be quite restrictive and the statement of the theorem is not precise enough. Please find my detailed comments below: 1) There are a lot of assumptions on homogeneity among agents. For example, the transition probability and the reward of each agent only depend on his/her type. The complexity of the problem depends on the number of the types instead of the number of agents. If this paper aims to study the mean-field limit, these assumptions are acceptable. Instead, this paper focuses on scenarios with finite number of players and these assumptions seem to be a bit restrictive 2) The term “pure strategy” in the paper seems to be misleading. In the definition of admissible policy set $\mathcal{X}$ (Eqn 1), $\Delta(A)$ is defined as the distribution over the action space. This implies that the admissible policies considered are “mixed” strategies instead of “pure” strategies. The authors did not define what they mean by “pure strategy” properly. 3) Theorem 1 is not stated rigorously. How big is the “finite number of steps”? What is the dependency on model parameters? What is the stopping criteria for the endpoint? 4) Is it possible to consider the dependency on other agents’ states in the reward?

Correctness: Statement of Theorem 1 is not mathematically rigorous enough.

Clarity: The paper is well-structured. The presentation can be improved by defining notations more clearly and making the statements more precisely.

Relation to Prior Work: Yes.

Reproducibility: Yes

Additional Feedback: Minor comments: 1) ‘’Partial observations” in the title is a bit misleading. It is better to describe the problem as learning with “local” policies. 2) The introduction of both $\Lambda_i$ and $\lambda_i$ are not well motivated. It seems that keeping one of them is enough to introduce the “types” of agents. Also, are $\Lambda_i$ defined on the same space for all i? Please define the notations more rigorously. 3) The statement in lines 113-114 is not accurate. For all randomized policies, the agents can behave differently even if they sample from the same policies. Typically this is not considered as heterogeneity of behaviors. This is rather the nature of randomized policy (or mixed strategy in the game literature). ================= [Further Feedback] Previously, my major concerns were on the assumption of agent heterogeneity and the rigorousness of the theorem statement. After reading the response from the authors and the reviews from other referees, I think my concerns are largely addressed. However, I still think Theorem 1 can be presented more rigorously. Therefore, I will raise my score to 6.


Review 4

Summary and Contributions: The paper presents the concept of shared equilibrium in certain kinds of multi agent stochastic games with a restricted form of partial observability. The formalism includes the notion of supertypes (different distributions of agents) and types (where each agents is given a true type each episode). The agent's type influences the rewards available as does the joint state of the system and joint action over all agents. One key constraint is that all agents of the same type follow the same policy from an egocentric perspective (where they themselves are the focal agent and all other agents are interchangeable). They define a policy gradient approach for individual agents, also present a higher order learning rule that shifts the distribution over supertypes at a slower timescale. This latter mechanism (in essence) allows the system to reach equilibrium for each given supertype profile before incrementally changing that to a different distribution over supertypes, and so can be used to train the supertypes to reflect observed characteristics in real systems or to guarantee fairness/liveness/etc properties for game design purposes. The paper is (on the whole) very clearly written and appropriately places itself within the literature. It motivates the problem domain well and constructs theoretical definitions, theorems and proofs clearly and explicitly. In some places, the text becomes a little discursive on formal issues and this does affect readability (see clarity). Nonetheless the overall theoretical picture is clear. The experimental description and interpretation was less clear to me (see clarity). Nonetheless, it is clear that some of the claimed properties are shown in some experiments. On the strength of the theoretical part of the paper I am recommending a weak accept, but I would request that the authors revisit the mentioned sections to try to improve their clarity.

Strengths: The paper clearly motivates the problem domain, defines a clear and formal description of an equilibrium concept (or concepts really), proposes a two-time scale algorithm with a clear description of the assumptions and interleaves this with a clear, concise intuition that facilitates reading. In my opinion it is a very high quality paper in this regard.

Weaknesses: There are some issues of clarity, and in particular in the experimental section (as described elsewhere - see summary and clarity sections). In particular, the experiments could more clearly support the theoretical contribution. Nonetheless, this is more a matter of communication than of appropriate choice of experiment.

Correctness: I was able to give a high level check of definitions, theorems and proofs, excepting Proposition 1 and the associated proof, where I found the overloading of variable names and the dual name of one of these to be a little difficult to navigate.

Clarity: I am not sure I fully understand the notation in Proposition 1 and its proof. When you say: $\nabla_{\theta_1} V_{\Lambda_i}(\pi_\theta, \pi_\theta)$ it is said that this means the gradient of with respect to the first of the two $\theta$s but really these are two independent parameters. I don't know what to suggest as a better notation though. In the description of the supertype learning, the supertypes were previously defined as discrete types, but this relies on their representations as mixtures of types (in some way). I guess I was originally expecting to read about how agents were shifted between fixed supertypes rather than the supertype itself moving, which may have been influenced by the original introduction of supertypes. Moreover, the grouping of agents under a given supertype allows a more compact/consistent learning at this higher level but this could be formalised more clearly. The experimental description and interpretation was less clear to me. Some of this is pushed to the supplementary material but a high level description might be considered in the paper. The text in the images is far too small too. More importantly, I struggled to pull a meaningful narrative out of what was being tested and how to interpret the resulting plots.

Relation to Prior Work: The paper relates well to prior work including the relevant multi-agent learning literature and two timescale learning.

Reproducibility: Yes

Additional Feedback: On the whole the language was very clear. I spotted one typo: [line 236] typically much less distinct supertypes -> typically many fewer distinct supertypes OR typically much smaller number of distinct supertypes ## Post-rebuttal After reading the authors' rebuttal, the other reviews and taking part in reviewer discussions I am still of the opinion that this is a good paper and worthy of acceptance. The explanations for the experiments in the rebuttal made things a bit clearer, but I still think this is an area where the paper could be improved. One or two of the other reviewers also pointed out that the class of games defined by the extended transitivity assumption isn't entirely clear. I think this is a good point and the description in the rebuttal only goes some way to addressing this. The space-invaders example is maybe a little limited in scope. Given all this, I have kept my original recommendation of marginally above the threshold.

[Author Response · NeurIPS 2020]

We thank all reviewers for the quality of their reviews. We have grouped in the 1st paragraph the common questions.

**All reviewers**. **Clarification of experiment goals**. The goal of the experiments is threefold: 1) Check that the
calibration targets are reached: for each experiment $j$ we have $n_j$ targets that we want to match simultaneously (table
3 in appendix). We present some of these targets in Fig. 2, and targets for all experiments are presented in appendix
Fig. 3-7. The calibrator agent's reward in Fig. 1 (and appendix Fig. 2) quantifies precisely the fit (1=perfect fit, cf.
appendix B2). 2) We check empirically in Fig. 1 that the agents rewards converge (supertype 1 and 2), indicating that
equilibrium is reached. 3) We check that our algorithm smoothly varies the parameters to be calibrated (supertypes)
in Fig. 3 and appendix Fig. 8-12, hence preventing potential reward divergence observed for Bayesian optimization
in Fig. 1 (experiment 4). **Extended transitivity assumption**. Our goal is to understand the implications of using a
shared policy and in particular the gradient update (3). Due to (4), (3) is in fact self-play for a certain game (Def. 1),
which implies that you need a class of games for which self-play converges, which we know is a strong requirement
[3]: there, it is said that self-play works for "transitive games", hence we generalize the transitivity assumption in [3]
from zero sum to general sum 2-player games. We do not have yet a full picture of the class of extended transitive
games, but we provide intuition behind the concept with the following example: assume you play SpaceInvaders but
with $n$ players on the screen. At the start, players are dummy and miss enemies. Then, one player becomes smarter and
finds a way to hit enemies. Transferring that knowledge to other players will make player 1 worse-off (lower score
due to other players hitting enemies), but still better-off than at the beginning where he/she was missing enemies: this
game is extended transitive. We believe that extended transitive games are those where there is some reward in the
outside world (enemies in SpaceInvaders, customer transactions in our paper) that players can collect by learning *game*
*skill*. **Potential and Bayesian games**. Extended transitive games are not potential games since there is no common
potential that gets increased whenever a player increases its utility: indeed, in assumption 1, the second step in moving
$(x,x) \to (y,x) \to (y,y)$ may not be an improvement for player 1, however it can be seen as a kind of "2-step potential
game" since the utility gets improved over 2 steps based on single player improvement. Our game associated to $\widehat{V}$
implicitly involves types (inside the expectation) as in Bayesian games, however because of the trick of going from $n$ to
2 players (the latter are 2 abstract players in that they are not part of the $n$ agents), the 2nd argument of the function
$\widehat{V}(\cdot, \cdot)$ actually ties out $n-1$ agents together with a policy in $\mathcal{X}$, hence we chose to introduce the terminology "shared
equilibrium" since the shared nature of the policy is rooted in the definition of the game $\widehat{V}$.

**R1**. We do not claim that the game where each player $i$ receives a utility of $V_i(\pi_i, \pi_{-i}; \Lambda_i, \Lambda_{-i})$ is symmetric: indeed
as you correctly say, it is not, because you would have to permute the supertypes $\Lambda$ too if you permute the $\pi$'s. We claim
that the 2-player game of Def. 1 with payoff $\widehat{V}$ is symmetric: here we think the confusion comes from our definition of
the word "payoff" in L159. Any 2 player symmetric game where $u_i(\pi_1, \pi_2)$ is the utility received by player $i$ satisfies
by symmetry $u_1(\pi_1, \pi_2) = g(\pi_1, \pi_2)$ and $u_2(\pi_1, \pi_2) = g(\pi_2, \pi_1)$ for some $g$ which we define as *payoff* in L159. The
game of Def. 1 is symmetric by construction since we define it as the symmetric game associated to payoff $g := \widehat{V}$:
this is an abstract game in the sense that it has 2 abstract players that are not part of the $n$ agents, the 1st abstract player
chooses $\pi_1$ and gets $\widehat{V}(\pi_1, \pi_2)$ (cf. L161-167), which is the expected utility received by getting assigned a random
supertype using $\pi_1$ playing against all other agents using $\pi_2$, cf. (4). The 2nd abstract player receives $\widehat{V}(\pi_2, \pi_1)$. We
find insightful that the function $\widehat{V}$ emerges naturally out of the gradient update (3), due to (4); Existence + finiteness of
all self-play sequences used in the proof of Thm 2 is given by Lemma 2 and its proof.

**R2**. Benefits of formalizing the problem as RL: the conceptual problem being solved by the calibrator agent is an online
search in the supertype space in order to achieve calibration targets. In our approach, supertypes get updated smoothly
using properties of specific RL algos (PPO here). Further, one can allow the calibrator to sample $N$ consecutive (batches
of $B$) actions per policy update (instead of 1 in the vanilla version), thus evaluating the right direction to move to in the
supertype space based on a sequence of N observations/actions/rewards (we used $N = 3$ in our experiments, cf. L156
of appendix).

**R3**. Rewards $\mathcal{R}$ can depend on both other agents' states and actions, but not on who plays them: $\mathcal{R}$ has to be invariant
w.r.t. permutations of the other agents' states/actions, thus ensuring that the expected reward in (2) only depends on
$\Lambda_i$: we will add it to the revision. We do not study the mean-field limit but our finite player setting with supertypes
naturally allows to group agents under specific distributions of types, thus reducing the number of simulation parameters
while keeping heterogeneity in the MAS and allowing coherent scaling w.r.t. the number of agents: this is exploited
in sections 3 for calibration, and illustrated in the experiments (yes, $\Lambda_i$ is defined on the same space $\forall i$, cf. L97). In
L167-170 we defined a pure strategy as an element of the game's strategy space $\mathcal{X}$: this is consistent with functional
form games of [3]. Thm 1 gives insight on the nature of games for which we get convergence of self-play and states
that the endpoint is an $\epsilon-$Nash, where none of the 2 players can improve its utility of more than $\epsilon$ (stopping criterion).

**R4**. We will clarify the experiment section in the revision. Yes, mathematically, we mean the gradient of the function
$V_{\Lambda_i}$ with respect to its first variable, taken at the point $(\pi_\theta, \pi_\theta)$. We should clarify L97 that the space $\mathcal{S}^\Lambda$ is assumed to
be a subset of $\mathbb{R}^d$ as we do L234: since (groups of) agents are mapped to supertypes, "moving" a supertype in $\mathbb{R}^d$ as in
algo 1 precisely means agents shifting between "fixed" supertypes, but in a continuous space.

[Meta-Review · NeurIPS 2020]

The paper was refereed by 4 knowledgeable reviewers. All reviewers appreciated the contributions of the paper: - Formalization of self play and formal proof when it is guaranteed to converge - New algorithm for calibrating equilibria that is more effective than a naive use of BO. - Convincing results on a market agent scenario. The biggest concern that was discussed between the reviewers was the assumption of the extended transitivity. While this was addressed partially in the rebuttal, the authors should add a longer discussion in the paper for which games this assumption holds. However, after the discussion all reviewers agreed that the paper merits acceptance and I join this decision.